# Energy Storage and Electrocaloric Cooling Performance of Advanced Dielectrics

**DOI:** 10.3390/molecules26020481

**Published:** 2021-01-18

**Authors:** Yalong Zhang, Jie Chen, Huiyu Dan, Mudassar Maraj, Biaolin Peng, Wenhong Sun

**Affiliations:** 1Research Center for Optoelectronic Materials and Devices, School of Physical Science and Technology, Guangxi University, Nanning 530004, China; gnolaygnahz@163.com (Y.Z.); a709161389@163.com (J.C.); dhy2358@163.com (H.D.); mudassar@mail.ustc.edu.cn (M.M.); 2Guangxi Key Laboratory of Processing for Non-ferrous Metal and Featured Materials, Guangxi University, Nanning 530004, China

**Keywords:** dielectric materials, ferroelectric materials, energy storage density, electrocaloric effect, low-temperature polarization, GaN

## Abstract

Dielectric capacitors are widely used in pulse power systems, electric vehicles, aerospace, and defense technology as they are crucial for electronic components. Compact, lightweight, and diversified designs of electronic components are prerequisites for dielectric capacitors. Additionally, wide temperature stability and high energy storage density are equally important for dielectric materials. Ferroelectric materials, as special (spontaneously polarized) dielectric materials, show great potential in the field of pulse power capacitors having high dielectric breakdown strength, high polarization, low-temperature dependence and high energy storage density. The first part of this review briefly introduces dielectric materials and their energy storage performance. The second part elaborates performance characteristics of various ferroelectric materials in energy storage and refrigeration based on electrocaloric effect and briefly shed light on advantages and disadvantages of various common ferroelectric materials. Especially, we summarize the polarization effects of underlying substrates (such as GaN and Si) on the performance characteristics of ferroelectric materials. Finally, the review will be concluded with an outlook, discussing current challenges in the field of dielectric materials and prospective opportunities to assess their future progress.

## 1. Introduction

Owing to their insulating properties, dielectric materials are known as electrically insulating materials. According to the different dielectric materials, traditional dielectric capacitors can be divided into three categories. Firstly, there are polymer-based dielectric capacitors [1]. The dielectric materials of this type of capacitors are mainly polymer materials. Second, there are ceramic-based dielectric capacitors. The dielectric materials of such capacitors include multi-phase ceramics, glass ceramics, ceramic films, etc. Third is the polymer ceramic composite dielectric capacitors [2], using a variety of polymer and ceramic composite materials as storage medium.

With the development of science and technology, some dielectrics have been found to have special properties related to the polarization process. If the crystal dielectric does not have inversion center, it can produce polarization under the action of mechanical force, i.e., piezoelectricity. There is no inversion center, but there is a unique polar axis crystal that is different from other directions. The spontaneous polarization change occurs with temperature variation, which termed as pyroelectricity. When the spontaneous polarization dipole moment changes with the direction of the applied electric field, the relationship between its polarization and applied electric field is similar to the relationship between the magnetization intensity and magnetic field of the ferromagnetic material, therefore, it also has a hysteresis curve [3]. Materials with piezoelectricity, pyroelectricity, and ferroelectricity are collectively called functional materials. They are all important components of dielectrics and can be used as transducers for heat, sound, light, electricity, etc., and have extremely important uses in defense, detection, communications, and other important fields.

In this paper, the strong-field properties of various dielectric materials are introduced, focusing on the ferroelectric materials (with piezoelectric effect, pyroelectric effect and spontaneous polarization at the same time) and their uses in energy storage and electrocaloric refrigeration. The common methods of optimizing the energy storage performance of ferroelectric materials such as reducing sample thickness and electrode area; improving the density of materials; increasing the band gap; improving the insulation; implanting dead layer; annealing in oxygen; low-temperature-poling control the interaction of defect dipoles and nanodomains are discussed. The effect of doping some small radius elements with high valence ion at A site and doping metal oxides with large ionic radius at B site to stabilize B-O (A, B, and O are located at the vertex, body center, and face center of the perovskite structure, respectively) octahedron are also discussed [4]. Finally, the advantages and disadvantages of various common ferroelectric materials are discussed and the future of ferroelectric materials is analyzed. We also put forward a method of lead retention for the protection of lead-based ferroelectric films.

## 2. Classification of Dielectrics

### 2.1. Polymer-Based Dielectric Materials

The characteristics of polymers are high dielectric breakdown strength (>500 kV/cm) and low permittivity(1–20) [5]. There are many types of polymers and common polymer matrices include polyethylene, polyvinylidene fluoride, epoxy resin, and polyimide. The energy storage density of polymer membranes is relatively low due to the low permittivity.

Binary and ternary copolymers exhibit more excellent dielectric and energy storage properties than single polymers. Z. Zhang et al. [6] prepared a vinylidene fluoride/trifluoroethylene/chlorotrifluoroethylene (PVDF/TrFE/CTFE) polymer, and its breakdown strength was greater than 500 kV/mm and energy storage density was more than 13 J/cm^3^. X. Zhou et al. [7] prepared vinylidene fluoride/chlorotrifluoroethylene (PVDF/CTFE) copolymer with a breakdown strength of 600 kV/mm and an energy storage density of up to 25 J/cm^3^. XZ Chen et al. [8] further added a bisamide compound (denoted as BA) to (PVDF/CTFE) and the prepared polymer had an energy storage density of 22.5 J/cm^3^ at an electric field strength of 400 kV/mm (Figure 1). In comparison with ceramic materials, plasticity is the major breakthrough of polymer-based dielectric materials. In practical applications, polymer-based dielectric materials can be bent into complex shapes according to the requirements of electronic devices, thereby reducing the volume of the device.

Although the energy storage density of the multi-polymer medium is high at low operating temperature, consequently, high-temperature functioning is challenging for different purposes such as aerospace and underground drilling. For example, the currently used commercial polymer material, the biaxially oriented polypropylene (BOPP), can only be used at temperatures below 105 °C. To cope with these problems, two strategies seem feasible: (i) polymer dielectric capacitors must be equipped with a cooling system to reduce the ambient temperature in high-temperature applications; (ii) polymers with a high glass transition temperature (*T_g_*) are used in high-temperature capacitors to ensure the stability of dielectric properties at high temperatures. The former solution came with additional weight and volume, making the device heavy and difficult to achieve portability. The problem of the limited working temperature of the polymer dielectric material cannot be solved perfectly, which is an important research direction of polymer dielectric materials but that’s not in the scope of our current review.

### 2.2. Glass Ceramic-Based Dielectric Materials

Glass ceramics is also one of the common dielectric materials, which is a composite material that combines structural glass phase and crystal through high-temperature melting, forming and heat treatment. Glass ceramics have high breakdown strength due to the glass phase. Among them, the size of the crystal phase normally ranges from the nanometer level to the micrometer level and the number of crystals can reach 50% to 90%. If the original composition is adjusted to ferroelectric crystal phases, ferroelectric glass ceramics can be obtained, which have both high permittivity and high breakdown strength. According to the different ferroelectric phases, there are two common ferroelectric glass ceramics: firstly, Titanate ferroelectric glass ceramics, with perovskite phase such as barium strontium titanate. J. Wang et al. reported on the modification of BaCO_3_-SrCO_3_-TiO_2_-Al_2_O_3_-SiO_2_ glass ceramics and its optimal energy storage density reached 2.81 J/cm^3^ (Figure 2). W. Zhang et al. [9] adopted microwave-processed Ba_X_Sr_1-X_TiO_3_-(Ba-B-AI-Si-O) glass ceramics to increase the breakdown strength of the dielectric material to 1486 kV/cm and the energy storage density was 2.8 J/cm^3^. Secondly, the ferroelectric phase is generally a niobate ferroelectric glass ceramic with a tungsten bronze structure, such as barium strontium niobite. Another common niobate glass ceramic is based on potassium niobate and sodium niobate as the ferroelectric phase and its energy storage density is about 2 J/cm^3^ [10]. The main glitch in the study of glass ceramics is interface polarization. The residual polarization of the material is large and the energy loss during the discharge is serious enough. In response to this problem, high breakdown strength glass can be added to the ceramic. The doped glass forms a liquid phase at the grain boundaries, which not only improves the breakdown strength of the ceramic, but also contributes to the sintering of the ceramic.

The applications of the ceramic membranes are also very extensive. The energy storage density of the films is larger due to its greater dielectric breakdown strength. For instance, the breakdown strength of the barium strontium titanate film can reach 1.0 MV/cm and the energy storage density is 11 J/cm^3^ [2]. Ceramic dielectrics have become one of the ideal third-generation pulse dielectric materials because of their characteristics of coordinated and controllable performance, wide operating temperature range and long cycle life.

### 2.3. Polymer Ceramic Composite Dielectric Materials

Polymer ceramic composite materials are also an important class of dielectric materials. Their design includes a synergistic effect, as the introduction of inorganic nanoparticles in the polymer matrix supplements the high permittivity of inorganic materials with the high breakdown strength of polymers. Due to the addition of inorganic particles on the nanometer scale, the interface area is increased, and the exchange coupling is promoted through the dipole interface layer, thereby obtaining higher polarization and dielectric response. N. Guo et al. [11] prepared isotactic polypropylene BaTiO_3_/TiO_2_ composite material by the in-situ metal complex polymerization method. Its permittivity was about 6.1, breakdown strength was about 400 MV/m and the energy storage density was 9.4 J/cm^3^. In 2015, Q. Li et al. [12] studied the composite materials of boron nitride nanosheets and polymers (Figure 3). Their dielectric properties are stable in a wide temperature and frequency range (Figure 4), and are internally stable below 250 °C the breakdown strength reached 403 MV/m and the energy storage density was 1.8 J/cm^3^. This finding showed that organic materials can also work at high temperatures after modification, which was the breakthrough in electronic devices.

Another common composite materials are the conductive or semiconducting fillers, which on adding into the polymers, resulted as an increase in the permittivity of the dielectric material. Using this method, the permittivity of the composite material can be increased by tens or even hundreds of times. Y Zhang et al. [13] prepared the Zn-PVDF polymer composite material by flocculation method, which formed a 1.34 mm thick film by hot pressing, the permittivity and dissipation factor of this material at 1 kHz were 52 and 0.05, respectively. F. Fang et al. [14] prepared and studied PVDF/BaTiO_3_/Ag three-phase composite materials. When the filler content is 58%, the permittivity at 1 kHz was 160. ZM Dang et al. [15] prepared Ni-PVDF polymer composites using a relatively simple mixing and thermoforming method with a permittivity up to 400. M. Arbatti et al. [16] used copper-calcium titanate (CCTO) ceramics with similar properties as semiconductors as fillers into (PVDF-TrFE), forming of “sandwich” structure of hot-pressed composite materials with a permittivity of 610 at a frequency of 100 Hz (Figure 5).

In the field of polymer ceramic composite materials, just by selecting a suitable polymer matrix and ceramic filler, the dielectric and energy storage properties of a material can be well adjusted. This selection makes them superior to the polymer matrix. At the same time, optimizing the interface between polymer and ceramic ensure their “compatibility” and the uniformity of the composite film is also an important prerequisite for obtaining high breakdown strength. At present, there are still many unsolved mysteries in this field such as: (i) The addition of ceramics increases the dielectric constant of the material, but the dielectric breakdown strength of the material is reduced. Balance maintenance between the two to achieve the optimization of energy storage performance is the most critical issue. (ii) The addition of ceramic materials deteriorates the mechanical properties of the polymer and affects the processing properties of the film. (iii) The interfacial effect of polymer and ceramics can be used to improve the energy storage performance of composite materials [17], but the essential characteristics of the interface still need to be further studied.

### 2.4. Ceramic-Based Dielectric Materials

Extensive research has been done in ceramic dielectric materials by the scientific community. According to the different polarization curve properties, ceramic dielectrics can be divided into linear dielectrics ceramic, ferroelectric ceramic, relaxor ferroelectric ceramic, and antiferroelectric ceramic [18]. As shown in Figure 6, the four materials are introduced below.

#### Energy Storage Performance of Linear Dielectric

The polarization *P* of the linear dielectric material varies linearly with the electric field *E* (Figure 6a) and the permittivity does not change with the bias voltage field. The energy storage density of linear dielectric materials can be calculated by the following formula [19]:(1)$$W=\frac{1}{2}\varepsilon_{0}\varepsilon_{r}E^{2}$$
where *ε*_0_ is the vacuum permittivity, *ε_r_* is the relative permittivity of the material and *E* is the electric field strength. It can be seen from the formula that the energy storage density of the linear dielectric material is proportional to the square of the electric field strength.

SrTiO_3_ (STO) has a low Curie point and is paraelectric at room temperature, so it is generally classified as a linear dielectric ceramic. Research directions mainly include rare earth doping modification, preparation of new materials with high permittivity and low loss; adjustment of the preparation process, reduction of sintering temperature and increase of breakdown strength through the addition of glass. Z. Wang et al. [20] studied the effect of non-stoichiometric ratio (Sr/Ti = 0.994–1.004) on the structure and properties of SrTiO_3_ ceramics and found that when Sr/Ti = 0.996, the breakdown strength reached 283 kV/cm, the energy storage density is 1.21 J/cm^3^, which is a great improvement over stoichiometric STO (*W* = 0.7 J/cm^3^). L. Li et al. [21] added Ba/Cu(B_2_O_5_) to STO and found that 1.0 mol% amount performed best breakdown strength of 28.78 kV/mm and the energy storage density was 1.05 J/cm^3^. H. Y Zhou et al. [22] explored the performance of CaZr_X_Ti_1-X_O_3_, ceramics in which zirconium replaces titanium ions. When x = 0.4, the maximum energy storage density reached up to 2.7 J/cm^3^.

The advantage of the linear ceramic material is that its breakdown strength is higher than that of the ferroelectric material, the operation is stable and the loss is low. However, its permittivity is low and the energy storage density increases slowly with the electric field. The following will focus on ferroelectric materials.

## 3. Ferroelectric Materials

The ferroelectric material is a widely used multifunctional material having following main characteristics: (i) Along with spontaneous polarization, some special crystal structures generate electric dipoles as their centers of gravity of positive and negative charges are shifted in a specific direction. Spontaneous polarization exists in ferroelectrics without any generation. The polarization state of the ferroelectric will change under various external conditions. (ii) The presence of spontaneous polarization will increase the total energy of the material system to keep the system stable an electric domain structure will be formed inside the system. The electric dipoles in each electric domain will be arranged in the same direction and the electric dipoles in different electric domains will be arranged in different directions, the existence of electric domains shows that ferroelectrics have micro-ordered but macro-disordered structures. (iii) With ferroelectric loops, under the effect of an externally enhanced electric field, electric domain reversal will occur in the material.

The disappearance of electric domains in a direction different from that of the electric field and the nucleation of electric domains in the same direction as the electric field causes hysteresis loop *P*(*E*) macroscopically (Figure 7). At point O, no electric field is applied to the ferroelectric body, the electric domains inside the ferroelectric body are disordered and randomly distributed and they do not show polarization externally. In the O-A segment, as the electric field increases, the polarization increases slowly because the domain wall movement between the electric domains only occurs due to the small electric field. In the A-B segment, as the electric field continues to increase, the polarization increases rapidly, at this time new domains begin to form and the domains in parallel or near-parallel electric field direction gradually increase and replace the other-oriented domains. In the B-C segment, the polarization reaches the maximum value. At this time, most of the electric domains of the ferroelectric are parallel or close to the parallel electric field direction and the inversion of the electric domains tends to saturate. Point C is called the maximum polarization (*P_max_*). In the C-D segment, as the electric field decreases, the polarization begins to decrease. When the electric field intensity is zero the intersection point D of the hysteresis loop and the vertical axis is called the residual polarization (*P_r_*). When it reaches point F, the polarization at this time is zero, which is called the coercive field intensity (*E_c_*). The ferroelectric will form a saturated hysteresis loop after cycling between the maximum value of the positive and negative electric fields and the energy required to reverse the domain twice is the same as the area surrounded by the hysteresis loop [23].

### 3.1. The Paraelectric and Ferroelectric Phases of Ferroelectric Materials

According to the Landau–Ginzburg–Devonshire phase transition theory, the order parameter of the ferroelectric is the spontaneous polarization (*P*). The free energy of the ferroelectric can be expanded as follows [25]:(2)$$F=F_{0}+\frac{1}{2}\alpha P^{2}+\frac{1}{4}\beta P^{4}+\frac{1}{6}\gamma P^{6}$$
where *α*, *β,* and *γ* are the Landau-type expansion coefficients. Here, the effects due to charge carriers, external electric field, applied stress, etc. are ignored. The polarization is assumed to be uniform. To reflect the phase change, *α* has the following form:(3)$$\alpha =\alpha_{0}\left(T-T_{c}\right)$$

In order to describe the second-order phase transition, *γ* in expression (3) needs to be taken as zero (*γ* = 0). In this way, free energy takes the following form:(4)$$F=F_{0}+\frac{1}{2}\alpha P^{2}+\frac{1}{4}\beta P^{4}$$

Equilibrium conditions for minimum output of free energy (first derivative equal to zero or second derivative greater than zero) are:(5)$$\frac{\partial F}{\partial P}=\alpha P+\beta P^{3}=0$$
(6)$$\frac{\partial^{2}F}{\partial P^{2}}=\alpha +3\beta P^{2}>0$$

Equation (5) has two solutions: *P = 0* which corresponds to the paraelectric phase; and $P^{2}\;=\;-\alpha /\beta$ which corresponds to the spontaneous polarizability of the ferroelectric phase. The solution of Equation (6) is: *α* > 0, which corresponds to paraelectric phase, and *α* < 0, which corresponds to ferroelectric phase.

### 3.2. Energy Storage Performance of Ferroelectric Ceramics

The permittivity of relaxor ferroelectrics and antiferroelectric changes with the electric field [26]. The energy storage density (*W*) and energy storage efficiency (*η*) are calculated as follows [26]:(7)$$W=\int_{P_{r}}^{P_{max}}EdP$$
(8)$$\eta =\frac{W_{energy}}{W_{energy}+W_{loss}}\times 100\%$$
where *E* and *P* are the applied electric field and polarization, respectively.

Currently, most widely studied ferroelectric ceramics are BaTiO_3_ ceramics, which have a large polarization and a large storage energy density. The most common modification method for barium titanate ceramics is the doping of oxides. W. Deng [27] used three kinds of mixed oxides to greatly improve the breakdown strength of BaTiO_3_ ceramics so that the energy storage density increased nearly twice that of pure BaTiO_3_. T. Wang et al. [28] used ZnNb_2_O_6_ doping to improve the breakdown strength of BaTiO_3_. The incorporation of a small amount of glass also improved the breakdown strength of barium titanate, thereby achieving the purpose of enhanced energy storage density [29].

Research on energy storage applications of ferroelectric (Ba,Sr)TiO_3_ (BST) ceramics in the paraelectric state is also a popular direction which includes oxide doping modification and the effect of grain size change on energy storage performance. The energy storage density of the BST-based ceramic block is greatly improved compared to the BTO-based ceramic block. G. Dong et al. [30] studied the structure and properties of ZnO-doped Ba_0.3_Sr_0.7_TiO_3_ and found that when the doping amount was 1.6 wt%, it had the maximum energy storage density of 3.9 J/cm^3^ at a breakdown strength of 40 kV/mm. Z. Song [31] prepared Ba_0.4_Sr_0.6_TiO_3_ with different grain sizes of 0.5–5.6 μm by controlling the sintering system and found that the material with the smallest grain size (0.5 μm) had the highest breakdown strength of 243 kV/cm with energy storage density of 1.28 J/cm^3^.

Although, ferroelectric ceramics have the advantages of high permittivity and good reliability, the dissipation factor is also large and the energy lost during the discharge is large. Additionally, the micro-cracks caused by electrostriction will reduce the breakdown strength, which is not conducive to energy storage.

### 3.3. Energy Storage Performance of Relaxor Ferroelectrics

Relaxor ferroelectrics, which were discovered in the 1950s, were advantages for energy storage applications because of their large polarization and narrow *P*–*E* curves, as shown in Figure 6c. Due to their strong thermal insensitivity of dielectric permittivities, they have been gradually paid attention and turned out to be a major class of important dielectric materials. The chemical composition of relaxor ferroelectrics is extremely complex and microscopically inhomogeneous.

Relaxor ferroelectrics contain a wide range of material systems among which lead-free systems are mainly solid solutions of BaTiO_3_ and some other perovskites. For example, (0.6BaTiO_3_-0.4BiScO_3_)_1−x_-(K_0.5_Bi_0.5_TiO_3_)_x_ ceramic solid solution was prepared by hot pressing sintering along with energy storage density of 4.0 J/cm^3^ at a bias voltage of 200 kV/cm [32]. The 0.7BaTiO_3_-0.3BiScO_3_ based capacitor, prepared by the printing method had a breakdown strength of 730 kV/mm, an energy storage density of 6.1 J/cm^3^ and the energy storage density was found to be very good and stable in the range of 0~300 °C [33]. The layered structure of Bi(Mg, X)O_3_ (X = Ti, Nb) with complex ions of BaTiO_3_ and B site forms a solid solution, which can also improve the energy storage of the material to a certain extent of performance [34,35]. Lead-based relaxor ferroelectrics mainly include Pb(Mg_1/3_Nb_2/3_)O_3_-PbTiO_3_, (Pb,La)(Zr,Ti)O_3_, etc. K. Yao et al. [36] adopted the chemical solution method and prepared 0.462Pb(Zn_1/3_Nb_2/3_)O_3_-0.308Pb(Mg_1/3_Nb_2/3_)O_3_-0.23PbTiO_3_ relaxor ferroelectric thin film and their maximum energy storage density reached 15.8 J/cm^3^. X. Hao et al. [37] prepared (Pb_0.91_La_0.09_)(Zr_0.65_Ti_0.35_)O_3_ relaxor ferroelectric thin film by sol-gel method and the energy storage density was 28.7 J/cm^3^ under the electric field strength of 2040 kV/cm.

### 3.4. Energy Storage Performance of Antiferroelectrics

The polarization curve of antiferroelectric ceramics is an obvious “double hysteresis loop”, as shown in Figure 6d. The polarization increases linearly with the increase of the applied electric field. The larger the electric field, the greater the phase change. The *P*–*E* curve shows the characteristics of ferroelectrics.

Figure 8 compares the internal polarization processes of ferroelectric materials and antiferroelectric ones. In ferroelectric materials, adjacent dipoles within the same domain have the same polarization direction. The polarization direction is opposite in antiferroelectrics (Figure 8a), only when a sufficiently large electric field is applied and the dipoles can be realigned similar to the ferroelectric.

Antiferroelectric ceramics are mainly composed of lead zirconate-based solid solution and silver niobate-based solid solution. Bingyue Qu et al. [38] reported energy storage density of 2.48 J/cm^3^ at 295 kV/cm and high transparency in the visible spectra (~60% at 0.7µm) for 0.8(K_0.5_Na_0.5_)NbO_3_-0.2Sr(Sc_0.5_Nb_0.5_)O_3_ ceramics. They also reported *W_rec_* (2.6 J/cm^3^) at 400 kV/cm and energy storage efficiency (73.2%) were achieved at 0.8KNN-0.2SSN-0.5mol% ZnO ceramics [39]. Mingxing Zhou et al. [40] reported 0.8NaNbO_3_-0.2SrTiO_3_ ceramics sintered at 1250 °C with a high breakdown strength of 323 kV/cm, *W_rec_* = 3.02 J/cm^3^, *P_max_* = 34.487 µC/cm^2^, *η* = 80.7% at an applied electric field of 310 kV/cm. It also exhibited a high current density of 677 A/cm^2^, an ultrahigh power density of 23.7 MW/cm^3^ and a short duration of time (~225 ns). They also reported that a novel NaNbO_3_-based (Na_0.7_Bi_0.1_NbO_3_) ceramics sintered at 1150 °C demonstrated energy storage efficiency of 85.4% and energy storage density (4.03 J/cm^3^), *P_max_* = 37.51 µC/cm^2^ at 250 kV/cm simultaneously [41].

### 3.5. Energy Storage Properties of Common Ferroelectric Ceramic Systems

#### 3.5.1. AgNbO_3_ System

Lei Zhao et al. [42] fabricated Mn-doped AgNbO_3_ ceramics at x = 0.1 wt% by a conventional solid-state reaction in O_2_ atmosphere sintered at 1070 °C and found good thermal stability with recoverable energy storage density(*W_rec_* = 2.5–2.9 J/cm^3^), *η* of 57% over the temperature range of 20–180 °C at 150 kV/cm, 1Hz frequency and large antiferroelectric–ferroelectric electric field(*E_F_*)of 117 kV·cm^−1^ that revealed the potential of AgNbO_3_ as a promising lead-free ceramic for energy storage applications. Ye Tian et al. [43] prepared Ag_0.91_Bi_0.03_NbO_3_ with solid-state methods sintered at temperature between 1060 °C and 1090 °C in oxygen and obtained the energy storage density of 2.6 J·cm^−3^ and the high energy efficiency of 80%. Lei Zhao et al. [44] obtained high thermal stability AgNbO_3_-0.1 wt% WO_3_ ceramic when sintered at 1070 °C and got *W_rec_* = 3.25 J/cm^3^ at 20 °C with an applied electric field of 185 kV/cm. They also demonstrated a energy storage density (*W* ~ 4.2 J/cm^3^), *E_F_* = 189 kV/cm, breakdown electric field(*E_b_*) = 233 kV/cm, *P_max_* = 36.8 μC/cm^2^, *η* = 69% of Ag(Nb_0.85_Ta_0.15_)O_3_ ceramic when sintered at 1100 °C [45]. *E_F_* = 142 kV/cm, *P_max_* = 39.6 μC/cm^2^, *η* = 62%, *W_rec_* = 3.2 J/cm^3^ of Mn-doped Ag_0.97_La_0.01_NbO_3_ ceramics were obtained by Chenhong Xu et al. [46].

#### 3.5.2. BaTiO_3_ System

Qingyuan Hu et al. [34] obtained BT_0.88_-BMT_0.12_ and get *W_rec_* = 1.81 J/cm^3^ at room temperature and at 224 kV/cm electric field. Longwen Wu et al. [47] fabricated (1−x)BaTiO_3_-xBi(Zn_2/3_Nb_1/3_)O_3_ at x = 0.15 when sintered at 1175~1250 °C obtained energy efficiency of 93.5% at 131 kV·cm^−1^ and *W_rec_* = 0.79 J/cm^3^. Wen-Bo Li et al. [48] synthesized 0.88BT-0.12BLN ceramics sintered at 1220 °C with energy density of 2.032 J·cm^−3^ and charge-discharge efficiency of beyond 88% under 270 kV·cm^−1^. Qibin Yuan et al. [49] indicated that the 0.9BT-0.1BZZ ceramics were sintered at a temperature ranging from 1150 °C to 1300 °C have potential to become an excellent candidate for high energy density capacitors used over a wide temperature range and energy density of 2.46 J/cm^3^ is achieved at an electric field of 264 kV/cm near breakdown strength at ambient conditions. Mingxing Zhou et al. [50] reported 0.85BaTiO_3_-0.15Bi(Zn_1/2_Sn_1/2_)O_3_ sintered at 1150 °C with energy storage density (2.41 J/cm^3^) and energy storage efficiency of 91.6%, *P_max_* = 23.23 µC/cm^2^ at 230 kV/cm having excellent frequency stability (5–1000 Hz) and fatigue endurance (cycle number: 10^5^).

#### 3.5.3. Bi_0.5_Na_0.5_TiO_3_ System

Qi Xu et al. achieved *P_max_* = 45.1 µC/cm^2^, *W_rec_* = 0.71 J/cm^3^ at 7 kV/mm in 0.9BNTBT-0.1NN sintered at 1110 °C [51]. Haibo Yang et al. obtained 0.92BBNT-0.08NBN with *W_rec_* = 1.7 J/cm^3^ and the energy storage efficiency of 82% at 172 kV/cm [52]. They also synthesized 0.90LLBNTZ-0.10NBN, which was sintered at 1150 °C and got *W_rec_* = 2.04 J/cm^3^ at 178 kV/cm. The values of energy storage density and energy storage efficiency is 0.91 J/cm^3^ and 79.51%, respectively for the 0.90LLBNTZ-0.10NBN ceramic at 100 kV/cm and 90 °C. It can be concluded that the (1−x)LLBNTZ-xNBN ceramics are promising lead-free candidate materials for energy storage devices over a broad temperature range [53]. Zhang et al. prepared ((Bi_0.5_Na_0.5_TiO_3_)_0.88_-(BaTiO_3_)_0.12_)_1−x_-(LiNbO_3_)_x_ (x = 0.0, 0.01, 0.02, 0.03, 0.04, 0.05, 0.06 and 0.07) by solid-phase reaction ceramics [54] (Figure 9). BNT-BT ceramics achieved high thermal stability in the temperature range of approximately 80 K at x = 0.03; at x = 0.06, *η* ~49.3% and *W_rec_* ~0.665 J/cm^3^ at a breakdown field strength of 100 kV/cm (Figure 10).

### 3.6. Lead-Based and Lead-Free Ferroelectric Ceramics

Ferroelectric ceramics are usually divided into lead-based ceramics and lead-free ceramics. Lead free ceramics are widely accepted as environment-friendly materials but their properties are much worse than lead-based ceramics. Lead based ceramics usually exhibit better properties than lead-free ceramics due to their unique phase boundaries. Therefore, the efforts for better performance and to avoid the impact of lead pollution have become very important. How to avoid the environmental impact of lead pollution? As shown in the Figure 11, a lead-protective film is added between the substrate and the ferroelectric thin film and a lead-absorbing layer film is added between the top electrode and the ferroelectric thin film. Is it possible to apply this strategy to the protection of lead-based materials?

## 4. Design of a Lead-Free Ferroelectric Materials with High Energy Storage Density

Ferroelectric materials with large breakdown electric fields can be widely used in many strong electric field applications, such as solid-state electrocaloric refrigeration, energy storage, electro strain, piezoelectric transduction, pyroelectricity, etc. Dielectric capacitors with the high power density and excellent temperature stability are highly demanded in pulsed power systems. A shown by formula (6), high breakdown electric field (*E**_b_*), high *P**_max_*, and low *P**_r_* are preferred to achieve high energy storage density. To improve the dielectric breakdown strength of ferroelectric materials, the common strategies are: (1) reduce the sample thickness and electrode area [55,56]; (2) increase the density of the material [57]; (3) increase the width of the material forbidden band [58]; (4) increase the insulation of the material [59,60]; (5) implant dead layer [61,62]; (6) anneal in oxygen [63,64], and so on. Although these methods are very effective for systems that do not contain volatile elements such as BaTiO_3_, but for some systems rich in volatile elements, such as Bi, Pb, Na, K for example BiFeO_3_, PbZrO_3_, Na_0.5_K_0.5_NbO_3_, Na_0.5_Bi_0.5_TiO_3_, etc. These methods have limited effect. Therefore, Li et al. proposed a method to improve the dielectric breakdown strength and control the interaction between defect dipoles and nano domains by low-temperature-poling [65].

In recent years, the research focus is mainly on doping and adding new materials to improve energy storage performance of dielectric materials. The ways to increase the energy storage density through doping mainly include the following ideas: (1) doping high-valent ions at the A site to increase the cation vacancy concentration, in order to improve the *P_max_* and energy storage density [39,66]; (2) A-site substitution element with a small ion radius or doping the refractory metal oxide into the low-melting-point matrix to reduce the grain size of the solid solution. It will increase the dielectric breakdown strength (*E_b_*) and the energy storage density [4,67,68,69,70]; (3) stabilizing the B-O octahedron by doping the element with a small ion radius at the A site and a metal oxide with an insoluble and large ion radius at the B site increases energy storage density [71]. Examples of grain size reduction, as a result of A/B site substitution, aging, and low-temperature-poling are listed below.

### 4.1. Improving Energy Storage Performance by Reducing Grain Size

In the BaTiO_3_ ceramics, the relationship between *E_b_* and the grain size was reported to obey the following relationship [72]:(9)$$E_{b}=1/{\bar{G}}^{0.5}$$
where $\bar{G}$ refers to the average grain size in the bulk sample, *E_b_* designated as breakdown electric field. It is obvious from the formula that, the smaller the grain size, the higher will be the *E_b_*. The next few examples will verify the above conclusions. The *E_b_* of Ba_0.4_Sr_0.6_TiO_3_ ceramics increased from 114 to 243 kV·cm^−1^ by decreasing the grain size from 5.6 to 0.5 mm [32]. In the K_0.5_Na_0.5_NbO_3_–SrTiO_3_ ceramics, the *E_b_* enhanced significantly from 140 to 400 kV·cm^−1^ when the grain size is reduced from 2.26 to 0.28 mm [73].

Han et al. systematically studied the effect of Ba modification on phase structure, microstructure and electrical properties of AgNbO_3_. As shown in Figure 12 [74] inclusion of Ba^2+^ ion led to complex cell lattice evolution and significant refinement of grain size. With the addition of Ba element, the permittivity at room temperature increased significantly from 260~ to 350 °C. For the composition of Ag_0.96_Ba_0.02_NbO_3_, maximum energy efficiency of 46.5% and a recoverable energy density of 2.3 J/cm^3^ were obtained.

### 4.2. Improving Energy Storage Performance through Classical A, B, A/B Site Substitution

#### 4.2.1. Classical A Site Substitution

Luo et al. [75] obtained a *P_max_* of 39.6 μC·cm^−2^, an *E_b_* of 220 kV·cm^−1^ and a *W_rec_* of 3.55 J·cm^−3^ in Ag_0.92_Ca_0.04_NbO_3_ ceramics by conducting a solid-phase reaction at a temperature of 1350 °C. Compared with Ag^+^ (1.48 Å, CN = 12), Ca^2+^ (1.34 Å, CN = 12) possesses a much smaller ionic radius [3,76]. The substitution of Ag^+^ by bivalent Ca^2+^ cations created silver vacancies and enhanced polarization, stabilized AFE phase, increased *E_F_* and *E_A_*, decreased *E_F_*-*E_A_*, and reduced grain size to increase *E_b_* (Figure 13).

As shown in Figure 14, with the increase of x (Ca content) in CANx ceramics, *P*–*E* loops became thinner, *P_max_* increased, *E_b_* increased and reached a maximum value at x = 4, *P_r_* decreased continuously by varying x. Combined with Equation (7), it can be seen that *W_energy_* increased by increasing *P_max_* and *E_b_* while decreasing *P_r_*, thereby increasing energy storage efficiency(*η*).

#### 4.2.2. Classical B Site Substitution

Li et al. [45] prepared AgNb_x_Ta_1-x_O_3_ ceramics using solid-phase reaction method at 1100 °C. The energy storage density has thermal stability of ~4.2 J·cm^−3^ at 20~120 °C (Figure 15). Adding Ta to AgNbO_3_ caused a decrease in particle size and an increase in dielectric breakdown strength from 175 kV·cm^−1^ of pure AgNbO_3_ to 240 kV·cm^−1^.

#### 4.2.3. Classical A and B Site Substitution

Han et al. [40] synthesized (Sm_0.02_Ag_0.94_)(Nb_0.9_Ta_0.1_)O_3_ ceramics by solid-phase reaction method at 1100 °C with an energy storage density of 4.87 J/cm^3^ and it had good thermal stability in the temperature range of 20~140 °C. Substitution of Sm^3+^ ions at the A site to reduce the lattice volume and substitution of Ta^5+^ ions at the B site to stabilize the [Nb/TaO_6_] octahedral tilt angle and reduced the grain size. This improved the stability of antiferroelectric properties and energy storage density, as shown in Figure 16.

As shown in Figure 17a, Luo et al. prepared (Sm_x_Ag_1-3x_)(Nb_0.9_Ta_0.1_)O_3_(SANT) by the solid-phase method, with the increase of Sm^3+^ ion doping, dielectric breakdown strength *E_b_* showed a trend of increasing first and then decreasing (at 2%, it reached the maximum). As shown in Figure 17b, for the SANT system, as the doping content of Sm^3+^ ions continued to increase, the saturation polarization first increased and then decreased, reaching a maximum of 38.5 μC/cm^2^ at a content of 2% and decreased to 23.8 μC/cm^2^ at a content of 4%. The residual polarization decreased monotonically, indicating that the AFE phase was stabilized.

### 4.3. Improving Energy Storage Performance through Aging

Xu et al. prepared lead-free relaxor ferroelectric ceramics (1 − x)(BCT-BMT)-xBFO using a traditional solid-state reaction method to obtain an energy density of 0.52~0.58 J/cm^3^ and energy storage efficiency of 76%~82% and high thermal stability (change from 323 K to 423 K < 4%) thereby found that the dielectric breakdown strength of ceramics prepared at high doping levels (x = 0.06 and 0.07) decreased significantly (Figure 18). Therefore, it was proposed that to improve the energy storage performance, aging process was used and the breakdown strength was significantly increased from 10 kV/cm of the prepared sample to 100 kV/cm of the aged sample. Due to the recombination of electrons and holes in the materials, with the increase of aging time, the insulation properties of the aged samples are improved and the dielectric breakdown strength is enhanced [78].

### 4.4. Improving Energy Storage Performance by Low-Temperature Poling

The defect dipoles are disordered at *T*_1_ (room temperature). Due to the high activity of *T*_1_ defect dipole, the leakage current is large. When the electric field *E* is applied, the dielectric breakdown strength (DBS) of the films are very low. However, when the temperature drops to *T*_2_ (−196 °C), the leakage current is very small because of the decrease of defect dipole activity. When the electric field is applied, the disorder defect dipoles become order. As the temperature rises again to *T*_1_, the nanodomains are pinned by highly ordered defect dipoles at *T*_2_, the leakage current will be very small when the external electric field is applied. Therefore, the DBS of the films are greatly enhanced. We proposed a method to improve the dielectric breakdown electric field and control the interaction between defect dipoles and nano domains by low-temperature-poling (Figure 19). Through low-temperature-poling (Pb,La)(Zr,Sn,Ti)O_3_ thin films, the dielectric breakdown strength from 1286 kV/cm to 2000 kV/cm and its corresponding energy storage density was also enhanced from 16.6 J/cm^3^ to 31.2 J/cm^3^ (Figure 20) [65].

## 5. Thin-Film Energy Storage Performance

In recent years, the miniaturization of materials used in electronic devices restricted research on thin films. Compared to ceramic materials, thin films exhibited a smaller volume, better dielectric properties and ten times the energy storage density than former. Researchers use different physical and chemical methods to prepare ferroelectric thin films, such as pulsed laser deposition (PLD), radio frequency (RF) magnetron, sol-gel, chemical solution deposition (CSD), etc. For the same ferroelectric thin film, different dielectric properties and energy storage density can be obtained by changing the preferred orientation of the substrate layer and the thickness of the substrate.

### 5.1. Dielectric Properties of Several Lead-Free Ferroelectric Thin Films

G. Kang et al. [79] used sol-gel method for the preparation of 0.55Ba(Zr,Ti)O_3_(BZT)–0.45(Ba,Ca)TiO_3_(BCT) thin film and demonstrated a high permittivity of 2913 with a low dissipation factor of 0.06, remnant polarization of 15.8 μC/cm^2^ with coercive field of 58 kV/cm and an effective piezoelectric coefficient *d_33_* of 71.7 pm/V. A. Jalalian et al. [80] also used sol-gel method and obtained a very large piezoelectricity in 0.5Ba(Ti_0.80_Zr_0.20_)O_3_-0.5(Ba_0.70_Ca_0.30_)TiO_3_ lead-free nanostructures sintered as thin films (*d_33_* = 140 pm·V^−1^) and nanofibers (*d_33_* = 180 pm·V^−1^). Moreover, the 0.5BTZ-0.5BCT film obtained by this method showed a high permittivity (ε = 1756), dielectric tunability about 44%, *P_r_* around 17.2 μC/cm^2^ and *E_c_* about 39.0 kV/cm was put forwarded by Zengmei Wang et al. [81]. W. Li et al. [82] used the same method and focused the tunability and obtained tunability of 64% for the LaNiO_3_ (LNO)/Pt/Ti/SiO_2_/Si substrate. It was indicated that the (Ba,Ca)(Ti,Zr)O_3_ (BCZT) is a promising candidate for microelectromechanical systems (MEMs), non-volatile memories, photoelectric devices applications. They also found the BCZT thin films deposited on Pt/Ti/SiO_2_/Si exhibited higher Curie temperature (75 °C), better piezoelectric property (*d_33_* of 50 pm/V) and lower dissipation factor (0.02). Z. Cai et al. [83] synthesized 0.5BZT-0.5BCT thin films and the indicated that 800 °C annealed film showed potential application in MEMS devices with piezoelectric constant (*d_33_* = 85 pm/V), elastic modulus and hardness about 140.09 and 4.35 GPa. W.L. Li et al. [84] reported the (100)-oriented ferroelectric lead-free 0.45BZT–0.55BCT thin films on Pt(111)/Ti/SiO_2_/Si substrates by sol–gel method and LaNiO_3_ seed layer was introduced between the film and the substrate which had *d_33_* values around MPB and the piezoelectric coefficient was 131.5 pm/V. Y. Chen et al. [85] obtained BCZT thin film by sol-gel method and the annealing temperature was 550 °C. The X-ray Reflectivity(XRR) results indicated that low-temperature crystallization suppressed the interfacial diffusion and extended film applications by integration with a silicon substrate. W.L. Li et al. [86] synthesized Ba(Zr_0.2_Ti_0.8_)O_3_-0.5(Ba_0.7_Ca_0.3_)TiO_3_ by RF magnetron sputtering method and concluded that the nano -domain had deep effect on piezoelectricity of thin films for MPB composition. The *d_33_* was 258 pm/V and *ε* was 168 of 0.5BZT-0.5BCT thin film while *d_33_* was 122 pm/V and *ε* was 1046 of 0.5BZT-0.5BCT/LNO thin films. A comparatively large effective piezoelectric coefficient *d_33_* = 100.1 ± 5 pm/V of the 0.5BaZr_0.2_Ti_0.8_O_3_-0.5Ba_0.7_Ca_0.3_TiO_3_ (BZT-BCT) thin films were obtained by B. C. Luo et al. [87] used RF magnetron sputtering method and suggested its potential application for high-performance lead-free piezoelectric devices. Qianru Lin et al. [88] obtained the 0.5Ba(Zr_0.2_Ti_0.8_)O_3_–0.5(Ba_0.7_Ca_0.3_)TiO_3_ epitaxial thin films by the same method and get, *P_r_* = 14.5 μC/cm^2^, *d_33_* = 96 ± 5 pm/V. Q.R. Lin et al. [89] described the properties of 0.5Ba(Zr_0.2_Ti_0.8_)O_3_–0.5(Ba_0.7_Ca_0.3_)TiO_3_ thin films through PLD, i.e., *P_r_* = 17 μC/cm^2^, *d_33_* = 103 ± 5 pm/V. C. J. M. Daumont et al. [90] deposit a Ba_0.97_Ca_0.03_Ti_0.9625_Zr_0.0375_O_3_ thin films with a thickness of about 130 nm on IrO_2_/SiO_2_/Si substrates also used this method and tunability up to 60% at 400 kV·cm^−1^. A. Piorra et al. [91] prepared thin films of 0.5(Ba_0.7_Ca_0.3_TiO_3_) –0.5Ba(Zr_0.2_Ti_0.8_)O_3_ by PLD and got *d_33_* = 80 pm/V and *ε* = 1756.

### 5.2. Energy Storage of Ferroelectric Thin Films

Lead-free relaxor ferroelectric thin film 0.5(Ba_0.8_Ca_0.2_)TiO_3_-0.5Bi(Mg_0.5_Ti_0.5_)O_3_(BCT-BMT) was prepared on Pt(111)/TiO_x_/SiO_2_/Si substrate by sol-gel method [92]. The BCT-BMT films have large energy density of 51.7 J·cm^−3^ and a high power density of 1.15 × 10^10^ W·kg^−1^ at room temperature (Figure 21).

(Ba_0.904_Ca_0.096_)_0.9775_-xLa_0.015_(Zr_0.136_Ti_0.864_)O_3_ (La-BCZT) lead-free relaxor ferroelectric thin films were fabricated on the LaNiO_3_/Pt composite bottom electrode by sol-gel method [93]. At x = 0.0075, an energy storage density of *W*~15.5 J/cm^3^ was obtained and the energy storage efficiency was as high as *η*~93.7%. The film had high thermal stability and the changes in energy storage density from 20 °C to 260 °C and efficiency of about <5% at 20 °C to 140 °C were obtained (Figure 22). 

The energy storage density of the lead-free BiFeO_3_-BaTiO_3_-SrTiO_3_(BFBSTO) solid solution thin films designed by Nan et al. reached 112 J/cm^3^ and 80% energy efficiency [94] due to the regulation of cubic, rhombohedral and tetragonal coexistence of polycrystalline nano domains (Figure 23).

## 6. Electrocaloric (*EC*) Refrigeration

The refrigeration system usually uses the Carnot cycle principle for refrigeration. For example, common air conditioners in daily life require a large amount of refrigerant to be used in air conditioning refrigeration systems, such as Freon and chlorofluorocarbons. However, the leakage of refrigerant will destroy the ozone layer resulting in increased ultraviolet radiation to the ground and cause the greenhouse effect. Therefore, it has become one of the research topics of artificial refrigeration to study the replacement of refrigerant-based refrigeration technology. In electrocaloric (*EC*) effect, a polar material changes the polarization state due to the change of applied electric field, resulting in changing the adiabatic temperature or entropy [95]. The *EC* effect refrigeration is the inverse effect of the pyroelectric effect [96]. In contrast to refrigerant for cooling, the *EC* effect has many advantage such as, compact size, no pollution and long-term use. Theoretical research shows that the electrocaloric cooling has higher cooling efficiency than the traditional compressor refrigeration equipment. The Carnot coefficient of the *EC* effect refrigeration can reach more than 60%, while the Carnot coefficient of the traditional refrigeration system is less than 20%. The *EC* refrigeration is considered to be one of the advanced refrigeration technologies that replace refrigerant based refrigeration [97].

The principle of *EC* effect refrigeration is shown in Figure 24 The *EC* refrigeration is generally divided into four stages; the first stage (Figure 24a), When there is no electric field at room temperature, the internal dipoles are disordered and the entropy value is large. In the second stage (Figure 24b), the material is polarized under adiabatic conditions and an external electric field is applied to the left and right sides of the material. At this time, the arrangement of dipoles inside the material slowly ordered itself leads to decrement in internal entropy as a result the temperature of the material rises due to changes in the internal structure [98]. In the thermal insulation environment, in order to keep the total entropy constant, the material absorbs the external heat, which results in an external temperature decrease. In the third stage (Figure 24c), the applied electric field remains unchanged. In order to prevent the dipole from absorbing heat again after the electric field is removed, the heat transfer sheet contacts the heat of the material itself, so that the temperature of the material drops to room temperature [99]. In the fourth stage (Figure 24d), when the electric field is removed under adiabatic conditions, the internal dipole of the material changes from ordered to disordered arrangement and the internal entropy value gradually start increasing. The material absorbs heat from the outside. The temperature is lowered, which is the reason for the cooling effect.

There are generally two methods for measuring the value of the *EC* refrigeration, namely, the direct method and the indirect method. The direct method builds the test equipment for the researcher himself, which requires relatively accurate equipment and has errors. The indirect method is to use Maxwell’s equation (∂*P*/∂*T*)*_E_*= (∂*S*/∂*E*)*_T_* to calculate the *EC* refrigeration under the condition of approximately reversible adiabatic. Therefore, most researchers use indirect methods to test the value of the *EC* refrigeration. This experiment uses an indirect measurement method. The temperature change (∆*T*) and entropy change (∆*S*) of the *EC* refrigeration can be calculated by formula (10) and formula (11) [100], respectively.
(10)$$\Delta T=-\frac{1}{\rho }\;\int_{E_{1}}^{E_{2}}\left(T/C\right){\left(\partial P/\partial T\right)}_{E}dE$$
(11)$$\Delta S=-\frac{1}{\rho }\;\int_{E_{1}}^{E_{2}}{\left(\partial P/\partial T\right)}_{E}dE$$

In the formula, *ρ* and *C* represent the density and specific heat capacity, respectively. *P* refers to the *P_max_* under the action of the applied electric field strength *E*. According to the formula, if one wants to obtain large electrocaloric effect refrigeration, then one must obtain a large breakdown field strength and polarization change with temperature [101]. In order to realize the application requirements of the electrocaloric effect in ferroelectric material, scientists hope to obtain a material with large *EC* refrigeration in a wide temperature range around room temperature. According to reports, in 2006, Mischenko and others [102] published Pb(Zr_0.95_Ti_0.05_)O_3_ ferroelectric thin film that has a huge electrothermal effect (Δ*T* ~ 12 K). Peng et al. [103] published Pb_0.97_La_0.02_(Zr_0.65_Sn_0.3_Ti_0.05_)O_3_ relaxor antiferroelectric lead-containing thin film to achieve a large positive electrocaloric effect (Δ*T* ~ 20.7 K) in a wide operating temperature range (110 K). Wang et al. [104] prepared BaTiO_3_ ceramics modified by Ca^2+^ and Sn^4+^ using traditional solid-phase methods. The electric heating effect of Δ*T* ~ 0.63 K was obtained under the electric field intensity of 20 kV/cm.

Peng et al. [105] prepared Pb_0.8_Ba_0.2_ZrO_3_ (PBZ) relaxor ferroelectric thin films on Pt(111)/TiO_x_/SiO_2_/Si substrates using the sol-gel method. At 598 kV·cm^−1^, a Δ*T* = 45.3 K and Δ*S* = 46.9 J·K^−1^·kg^−1^ was obtained due to the electric field causing the nanoscale AFE to FE phase transition (Figure 25).

0.5(Ba_0.8_Ca_0.2_)TiO_3_-0.5Bi(Mg_0.5_Ti_0.5_)O_3_(BCT-BMT) relaxor ferroelectrics on Pt (111)/TiO_x_/SiO_2_/Si substrates were synthesized using the sol-gel method thin films (Figure 26). The negative electrocaloric (*EC*) effect of ∆*T* ~−42.5 K and ∆*S* ~−29.3 J·K^−1^·kg^−1^ was obtained at the electric field strength of 1632 kV/cm. Due to the applied electric field, a uniform rhombohedral phase transition occurs along the out-of-plane direction (Figure 27a). We proved this speculation by building a phase change model [92] (Figure 27b).

Pb_0.78_Ba_0.2_La_0.02_ZrO_3_ (PBLZ) relaxor ferroelectric thin films on different substrates were prepared by the sol-gel method [106]. The *EC* effect of PBLZ largely replaced the substrate materials and preferred orientation. Deposited on Pt(111)/TiO_x_/SiO_2_/Si(100)(Pt), LaNiO_3_/Pt(111)/TiO_x_/SiO_2_/Si(100) (LaNiO_3_/Pt), LaNiO_3_/n-type GaN (LaNiO_3_/n-GaN) and LaNiO_3_/p-type GaN (LaNiO_3_/p-GaN) substrates have a maximum Δ*T* of ~13.08 K, 16.46 K, 18.70 K and 14.64 K, respectively (Figure 28). The thin films deposited on LaNiO_3_/n-GaN and LaNiO_3_/p-GaN substrates obtained a negative *EC* effect in a wide temperature range (340~440 K).

GaN has wurtzite (hexagonal) structure at normal temperature and pressure. The lattice lacks inversion symmetry which creates spontaneous transformation and piezoelectric effect. The thin film deposited on LaNiO_3_/n-GaN substrate showed (100) preferred orientation and the thin film deposited on LaNiO_3_/p-GaN substrate showed weak (110) preferred orientation due to the unique spontaneous polarization of GaN orientation. The intensity of the OAFE (210) superlattice peak was stronger than that of the former, indicating that the orthorhombic ferroelectric phase accounted for a higher proportion of rhombohedral ferroelectric phase (Figure 29).

Nb-doped Pb_0.99_(Zr_0.65_Sn_0.3_Ti_0.05_)_0.98_O_3_ antiferroelectric thin films were prepared by sol-gel method and obtained a positive electrocaloric (*EC*) effect at 10 kHz (Δ*T* ~ 12.3 K, Δ*S* ~ 13.6 J·K^−1^·kg^−1^ at 293 K), a negative *EC* effect was obtained at 100 kHz (Δ*T* ~ −5.8 K and Δ*S* ~ −4.5 J·K^−1^·kg^−1^ at 425 K) (Figure 30) [107]. The large positive *EC* effects are ascribed to the co-coupling between the phase-transition (nanoscale tetragonal transferring into rhombohedral along the direction which is vertical to the surface of the thin film) & defect dipoles. The comparison of *EC* refrigeration of different thin films is described in Table 1.

## 7. Challenges and Prospects

### 7.1. Status and Challenges of Dielectric Materials

Dielectric materials have attracted research interest in the scientific community because of their use in many devices. Dielectric materials have fast charge and discharge speeds and high power density but their energy storage density is much lower than that of ordinary fuel cells and lithium batteries. Therefore, finding new dielectric materials with higher energy storage density are needed. Ferroelectric materials have grasped overwhelming attention due to their spontaneous transformation characteristics and high dielectric breakdown strength especially lead-based ferroelectric materials (unique MPB phase region and excellent performance). Years of development have made exciting progress hitherto, these materials have certain challenges. For instance, in PbZrO_3_ and PbMg_1/3_Nb_2/3_O_3_ systems, Pb and Mg volatilization problem and lead toxicity is vulnerable [117]. (Na_1/2_K_1/2_)NbO_3_ (KNN) based ceramics prepared by solid-state reaction need high sintering temperature and complex preparation process. The high volatility of K/Na in the high-temperature sintering process will lead to the decrease of the density of samples after sintering. The relative density had been reduced, which led to a reduction in its electrical performance and service life. (Na_1/2_Bi_1/2_)TiO_3_ (BNT) relaxor ferroelectric ceramics have low residual polarization (*P_r_*) and small *W_loss_* and showed excellent energy storage potential at low electric field strength but low dielectric breakdown strength (DBS) limits the enhancement of *W_rec_*. Moreover, due to the high-temperature volatility of Bi and Na, the overall performance of the material is affected. BiFeO_3_ (BFO) based ceramics have high leakage current and low voltage electric coefficient greatly limit its development. BaTiO_3_ (BTO) offers piezoelectric properties comparable to Pb-based ceramics (*d_33_* ~ 700 pC/N) through doping and other chemical modification. The preparation process is simple, e.g., BCZT system has MPB similar to PbZrO_3_ group. The huge electrostriction coefficient (*Q_33_* ~ 0.04–0.06 m^4^/C^2^), large dielectric breakdown strength, and large saturation polarization make it a part of the energy storage devices.

### 7.2. Future Prospects

With the miniaturization of electronic components, the future dielectric materials will be developed using thin-film technology, as the energy storage and electrocaloric performance of the some ferroelectric material films are found to be increased about ten times than that of bulk ceramics. Pulsed power systems place high demands on dielectric capacitors with the high power density, excellent temperature stability and relaxor ferroelectric/antiferroelectric materials as compared to ordinary dielectric materials. Stability has aroused extensive attention and achieved a rapid progress. The energy storage density of the lead-free BiFeO_3_-BaTiO_3_-SrTiO_3_ solid solution thin films designed by Nan et al. reached 112 J/cm^3^ and 80% energy efficiency [94]. Ferroelectric/antiferroelectric materials with large positive or negative electrocaloric (*EC*) effects have great potential in designing commercial refrigeration equipment. So far, negative *EC* has reached Δ*T* ~ 42.5 K [92] and the voltage equivalent to a few volts can instantly reach a temperature change of tens of Kelvin.

Today, the advancement in the fields of dielectric materials is still concentrated on lead-based ferroelectric materials but the lead bioaccumulation property make it vulnerable to human health as well as environment. Therefore, lead-free ferroelectric materials have gained attention but they still possesses lower performance in comparison to lead-based ferroelectric materials. As an effort, Xu et al. reported that a transparent lead-absorbing molecular film (containing phosphonic acid groups that are strongly bound to phosphorus) was used on the glass side of the front transparent conductive electrode. The polymer sheet blended with the mixture was placed between the metal electrode and the standard photovoltaic filler sheet resulting the lead retention efficiency up to 96% without effecting the performance of the device [118].

The lead retention strategy (Figure 11) combines a series of methods to improve the dielectric properties of ferroelectric materials, such as the interaction of low-temperature polarization control defect dipoles and nano-domains that we mentioned in Section 4. This verily describes the future of ferroelectric materials in the field of capacitors and electrocaloric effect refrigeration.

## Figures and Tables

**Figure 1 molecules-26-00481-f001:**
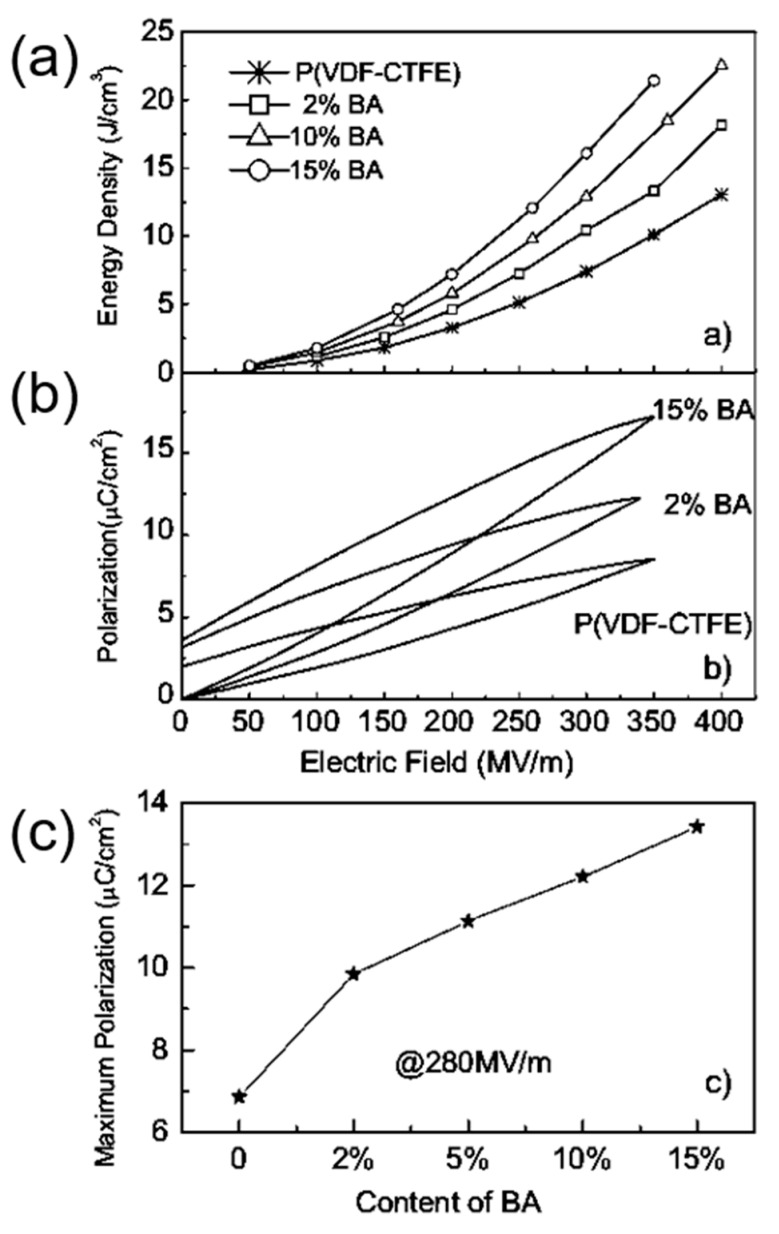
(**a**) Discharge energy density in (PVDF-CTFE) with different contents of cross-link agents. (**b**) Example of enhanced polarization in the cross-linked polymers shown by *P*–*E* loops at around 350 MV/m. (**c**). The maximum polarization of the copolymers at 280 MV/m. Reprinted with permission from [8]. Copyright 2011 WILEY-VCH Verlag GmbH & Co.

**Figure 2 molecules-26-00481-f002:**
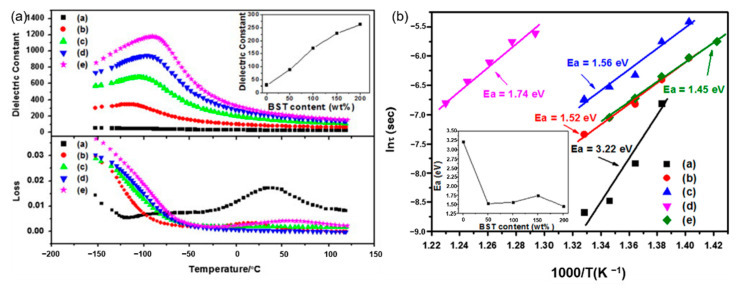
(**a**). Temperature dependence of permittivity and dissipation factor of (a) G-0 (b) G-50 (c) G-100 (d) G-150 and (e) G-200 (measured at 1 MHz). Inset: permittivity at room temperature of the samples. (**b**). Weibull plots of dielectric breakdown strength of (a) G-0 (b) G-50 (c) G-100 (d) G-150 and (e) G-200 (*E_a_*: average dielectric breakdown strength). Inset: *E_a_* of the samples. Reprinted with permission from [9]. Copyright 2013 Elsevier Ltd. and Techna Group S.r.l.

**Figure 3 molecules-26-00481-f003:**
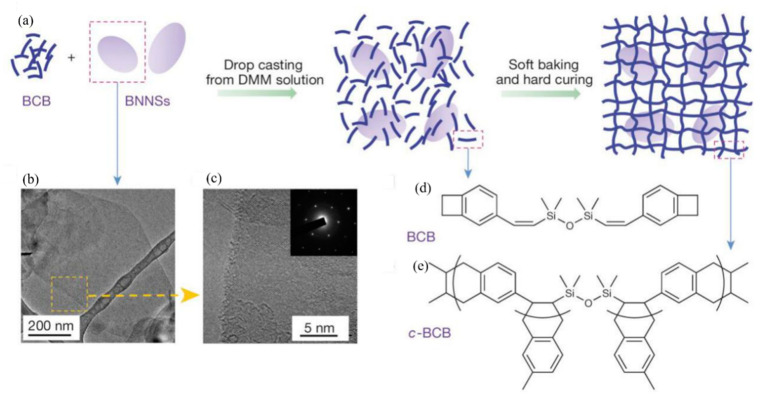
Material preparation and structures. (**a**) Schematic of the preparation of c-BCB/BNNS films. (**b**,**c**) Transmission electron microscopy (TEM) images of BNNSs exfoliated from h-BN powders. Inset of c is an electron-diffraction pattern of BNNSs, showing its hexagonal symmetry. (**d**) Chemical structure of the BCB monomer. (**e**) The repeating unit of c-BCB. Reprinted with permission from [12]. Copyright 2015, Springer Nature.

**Figure 4 molecules-26-00481-f004:**
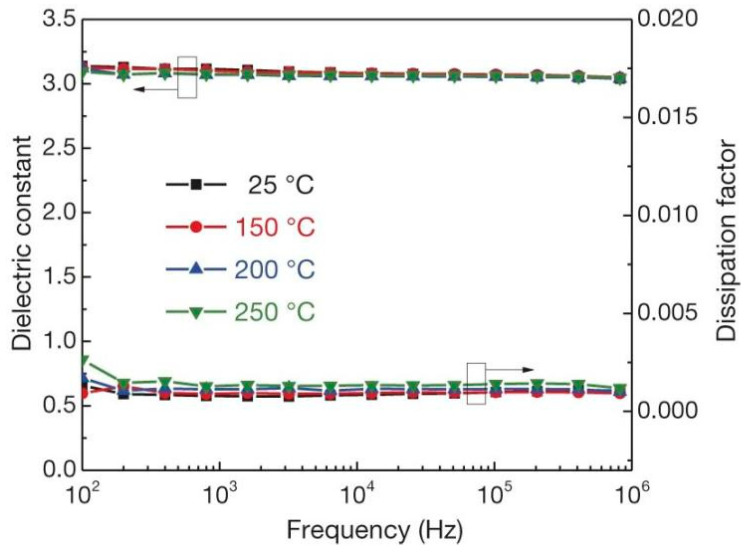
Frequency dependence of the permittivity and dissipation factor of c-BCB/BNNS with 10 vol% of BNNSs at different temperatures. Error bars show standard deviation. Reprinted with permission from [12]. Copyright 2015, Springer Nature.

**Figure 5 molecules-26-00481-f005:**
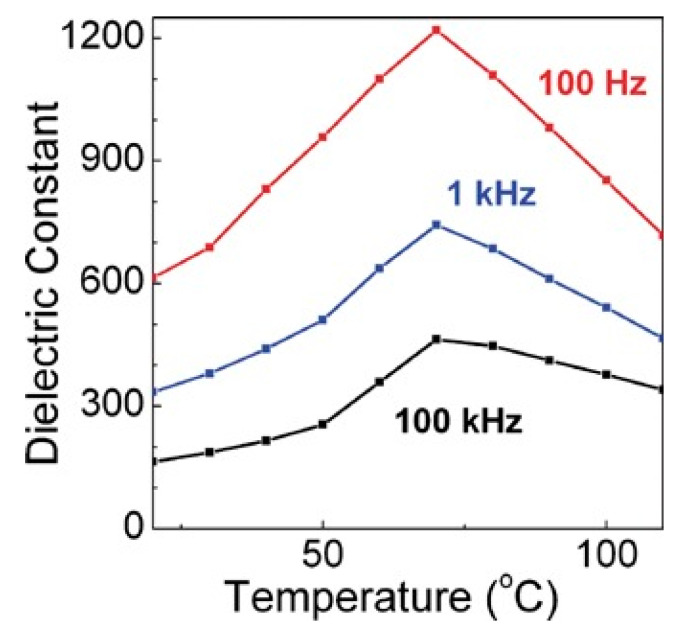
Permittivity at 100 Hz, 1 kHz and 100 kHz vs. temperature for hot-pressed composites. Reprinted with permission from [16]. Copyright 2007 WILEY-VCH Verlag GmbH & Co.

**Figure 6 molecules-26-00481-f006:**
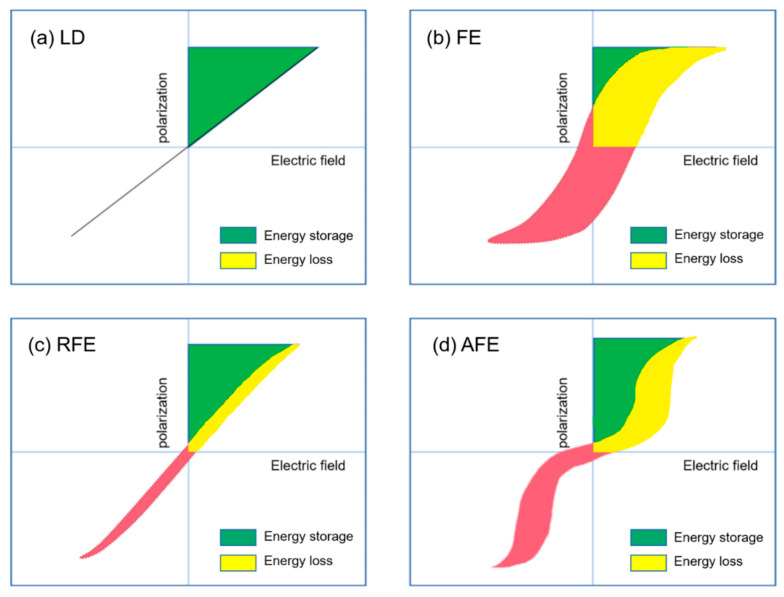
Hysteresis and corresponding energy storage density for (**a**) linear dielectrics, (**b**) ferroelectrics, (**c**) relaxor ferroelectrics and (**d**) antiferroelectrics. The green area represents energy density and the red area refers to the energy loss.

**Figure 7 molecules-26-00481-f007:**
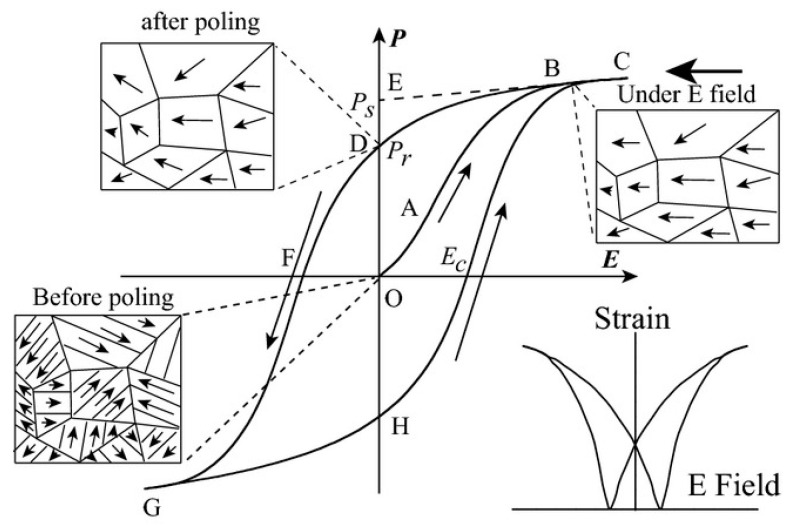
A typical hysteresis loop in ferroelectrics and strain-electric field curve. Reprinted with permission from [24]. Copyright 2013 The American Ceramic Society.

**Figure 8 molecules-26-00481-f008:**
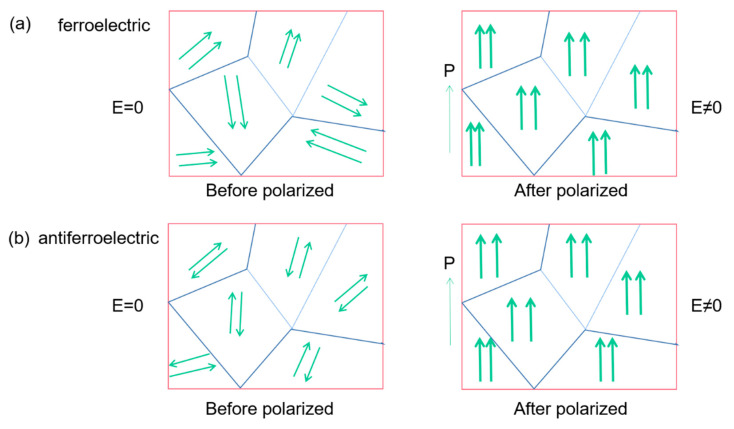
Schematic diagram of the polarization process in ferroelectrics and antiferroelectrics. The arrows indicate the polarization orientation. (**a**) Ferroelectric, the adjacent dipoles in the same domain are arranged in parallel. (**b**) Antiferroelectric, the adjacent dipoles in the same domain are arranged in reverse parallel.

**Figure 9 molecules-26-00481-f009:**
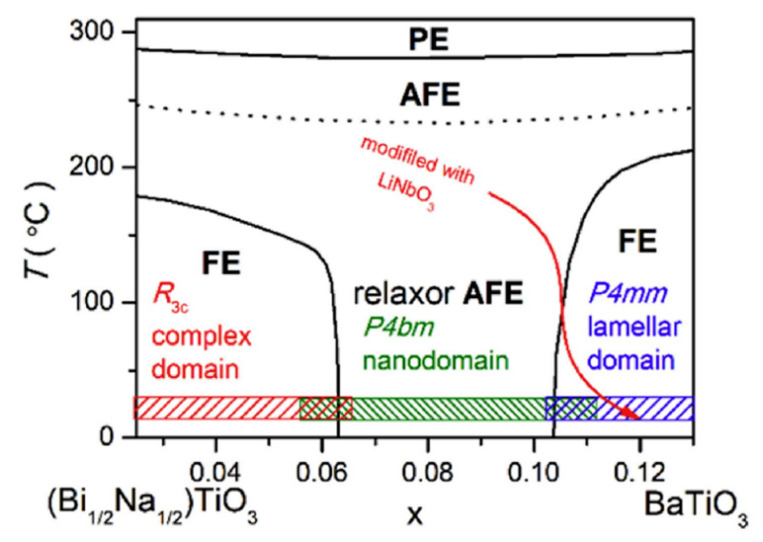
Phase diagram of the (1 − x)(Bi_1/2_Na_1/2_)TiO_3_-xBaTiO_3_ ceramics. PE, AFE and FE represent paraelectric, antiferroelectric and ferroelectric phase, respectively. Reprinted with permission from [54]. Copyright 2019 Elsevier Ltd. and Techna Group S.r.l.

**Figure 10 molecules-26-00481-f010:**
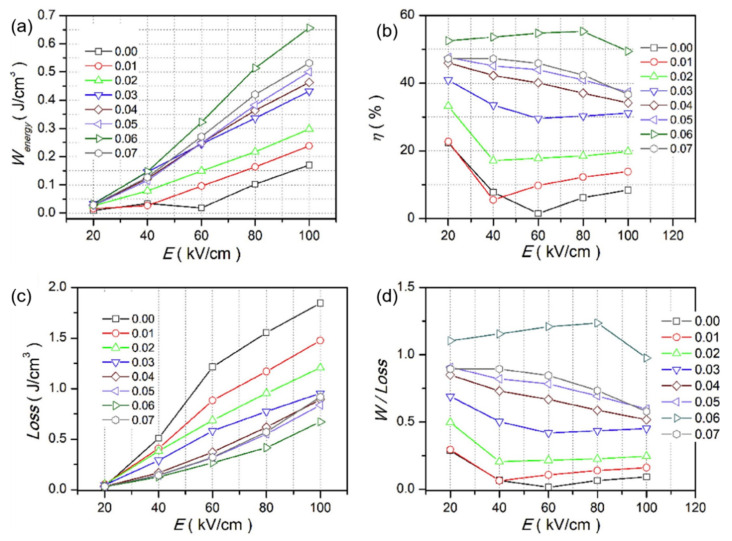
*W**_energy_*, *W**_loss_*, *η* and *W_energy_/W_loss_* of (1 − x)(BNT-12BT)-xLiNbO_3_ at selected electric fields. (**a**) *W**_energy_*, (**b**) *η*, (**c**) *W_loss_* (**d**) *W_energy_/W_loss_*. Reprinted with permission from [54]. Copyright 2019 Elsevier Ltd. and Techna Group S.r.l.

**Figure 11 molecules-26-00481-f011:**
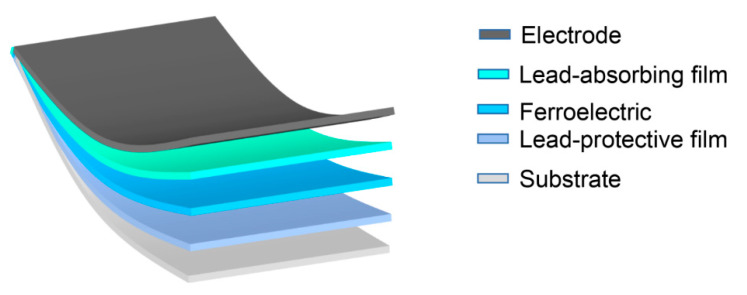
Imagination of Lead Sequestration Model Diagram. Composition structure (Electrode/Lead-absorbing film/Ferroelectric/Lead-protective film/Substrate).

**Figure 12 molecules-26-00481-f012:**
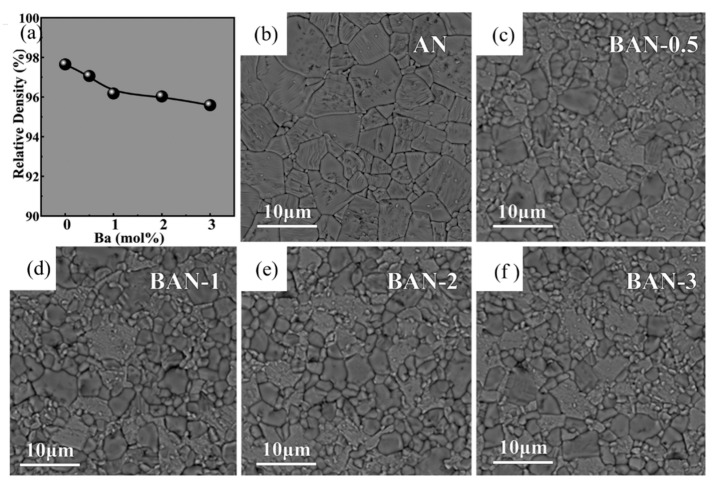
(**a**) The variation of relative density for the as-prepared Ba_x%_Ag_1-2x%_NbO_3_(BAN-*x*) ceramics, SEM images of (**b**) AN, (**c**) BAN-0.5, (**d**) BAN-1, (**e**) BAN-2 and (**f**) BAN-3 ceramics. Reprinted with permission from [74]. Copyright 2018 Elsevier Ltd. and Techna Group S.r.l.

**Figure 13 molecules-26-00481-f013:**
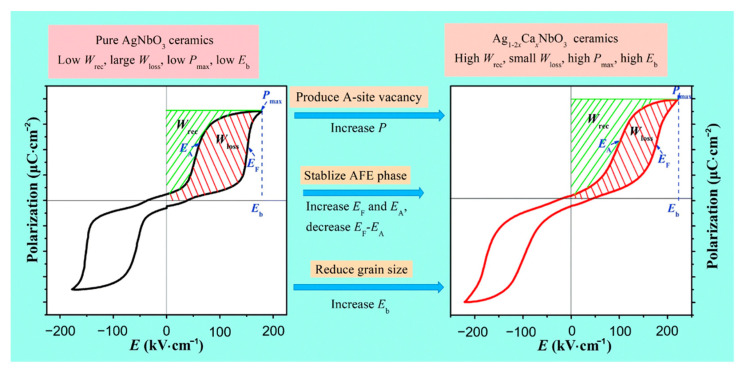
Schematic diagram showing the strategy for increasing energy storage performance. Reprinted with permission from [75]. Copyright 2019 Royal Society of Chemistry.

**Figure 14 molecules-26-00481-f014:**
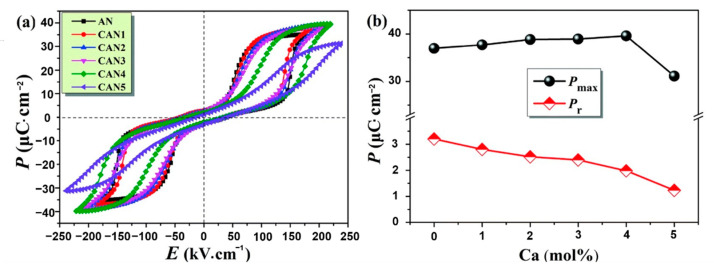
Ag_1__−2__x_Ca_x_NbO_3_ (CANx) ceramics. (**a**) *P–**E* loops; (**b**) P_max_ and P_r_. Reprinted with pemission from [75]. Copyright 2019 Royal Society of Chemistry.

**Figure 15 molecules-26-00481-f015:**
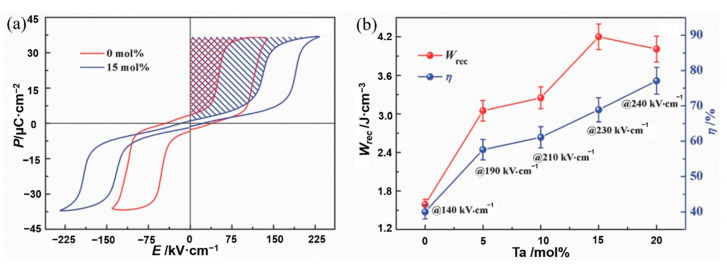
(**a**) Hysteresis loops of AgNbO_3_ and Ag(Nb_0.85_Ta_0.15_)O_3_ ceramic. (**b**) Energy storage performance of Ag(Nb_1−_*_x_*Ta*_x_*)O_3_ ceramics prior to their breakdown. Reprinted with permission from [45]. Copyright 2017 WILEY-VCH Verlag GmbH & Co.

**Figure 16 molecules-26-00481-f016:**
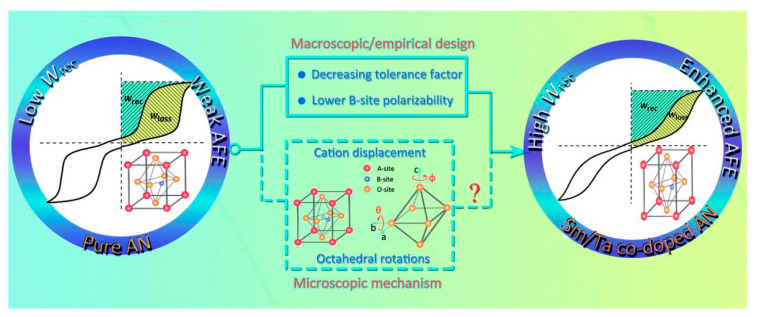
Schematic diagram of (Sm_0.02_Ag_0.94_)(Nb_0.9_Ta_0.1_)O_3_ enhancing *W**_rec_* by increasing the stability of AFE phase in this study. Reprinted with permission from [77]. Copyright 2019 Royal Society of Chemistry.

**Figure 17 molecules-26-00481-f017:**
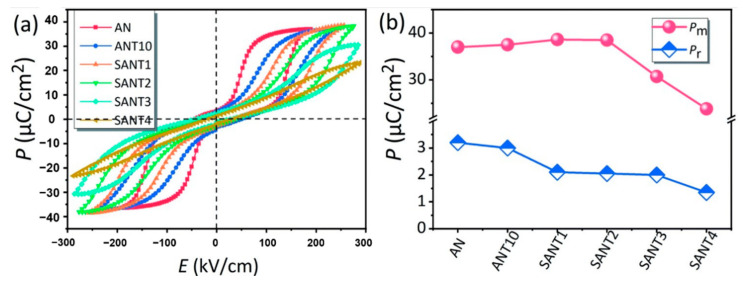
(**a**) plots the room-temperature *P*–*E* loops of the as-sintered ceramics measured at 1 Hz, (**b**) *P*_m_ and *P*_r_. Reprinted with permission from [77]. Copyright 2019 Royal Society of Chemistry.

**Figure 18 molecules-26-00481-f018:**
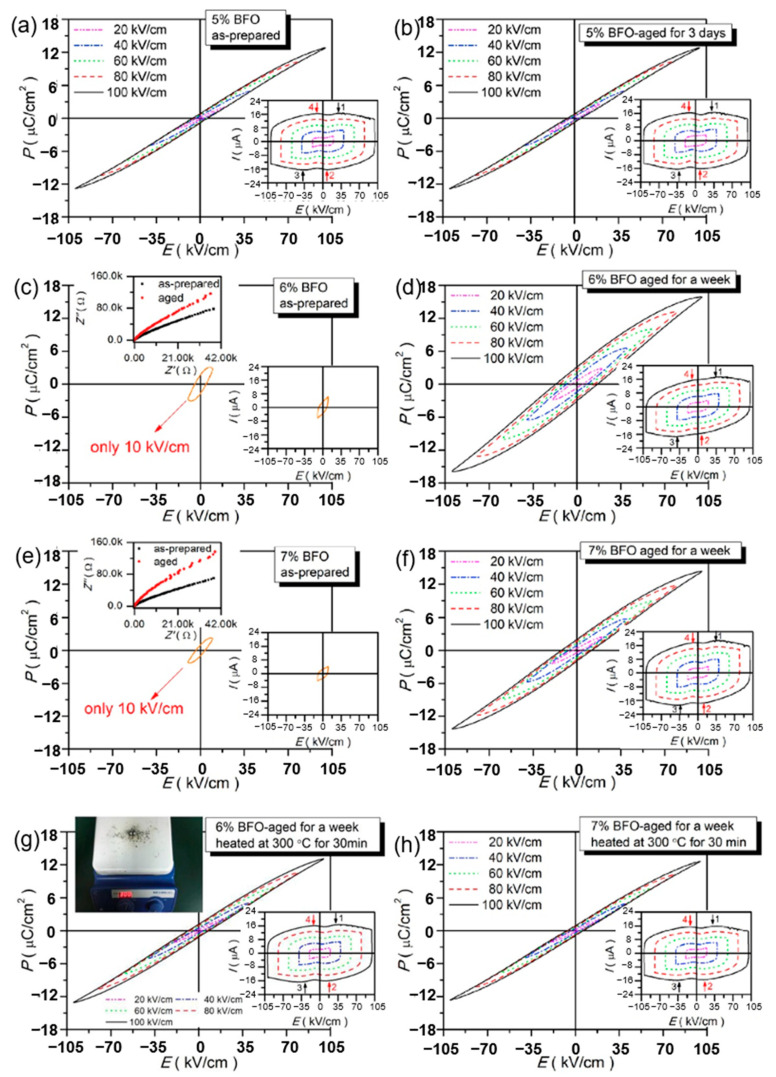
*P**-**E* loops of BFO-doped BCT-BMT-based as-prepared and aged ceramics. (**a**,**b**) x = 0.05; (**c**,**d**,**g**) x = 0.06; (**e**,**f**,**h**) x = 0.07. The left upper corners insets of (**c**,**e**): complex impedance plots of as-prepared and aged ceramics at x = 0.06 and 0.07 at room temperature. The left upper corners insets of (**g**): schematic diagram of aged samples at x = 0.06 and 0.07 heated at 300 °C for 30 min. The right lower corner insets of (**a**–**h**): *I-E* loops. Reprinted with permission from [78]. Copyright 2019 Elsevier B.V.

**Figure 19 molecules-26-00481-f019:**
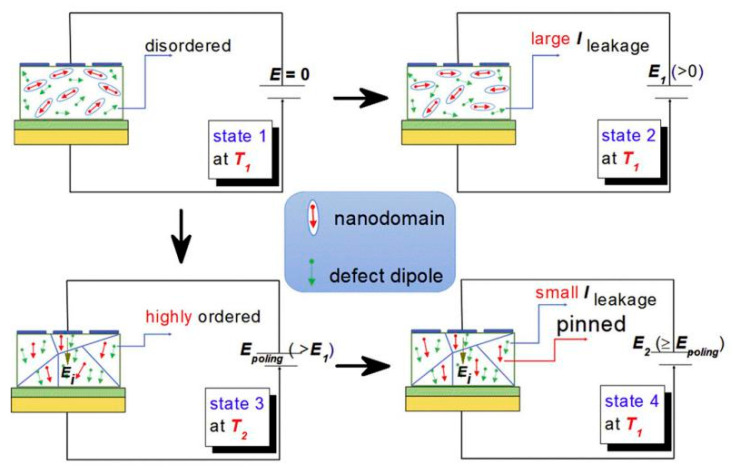
Model diagram of low temperature polarization increasing dielectric breakdown electric field. Reprinted with permission from [65]. Copyright 2020 Published by Elsevier Ltd.

**Figure 20 molecules-26-00481-f020:**
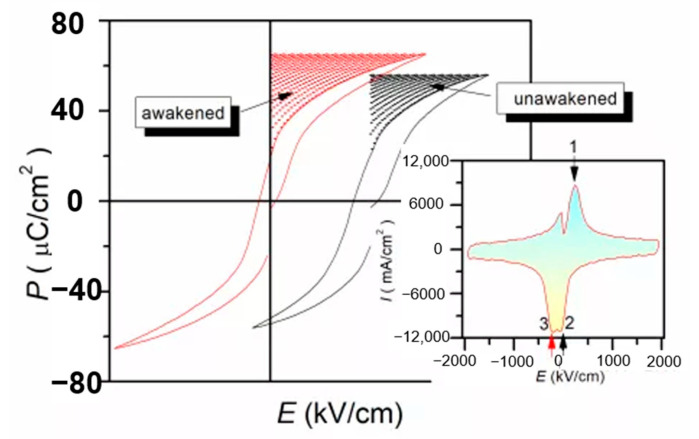
Comparison of energy storage density before polarization (wake up) and after polarization (wake up). Reprinted with permission from [65]. Copyright 2020 Published by Elsevier Ltd.

**Figure 21 molecules-26-00481-f021:**
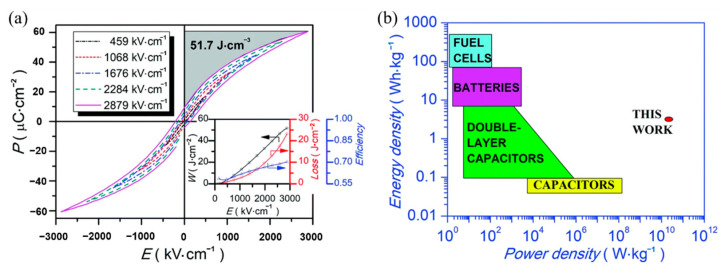
Energy density and power density measurements of the BCT–BMT thin film. (**a**) *P–**E* loops. Inset: *W_energy_*, *W_loss_* and Z versus E. (**b**) Energy density & power density diagram. Reprinted with permission from [92]. Copyright 2019 The Royal Society of Chemistry.

**Figure 22 molecules-26-00481-f022:**
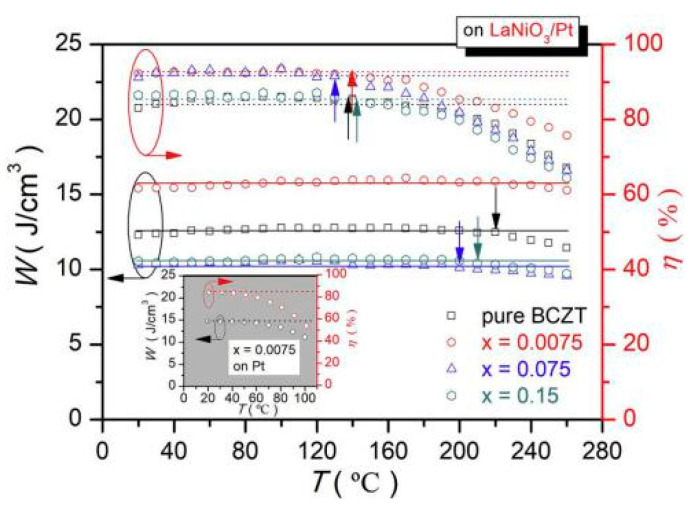
*W_(_**_T)_* and *η_(_**_T)_* of pure and La-doped BCZT thin films on LaNiO_3_/Pt Composite bottom electrodes. Reprinted with permission from [93]. Copyright 2019 Elsevier Ltd. and Techna Group S.r.l.

**Figure 23 molecules-26-00481-f023:**
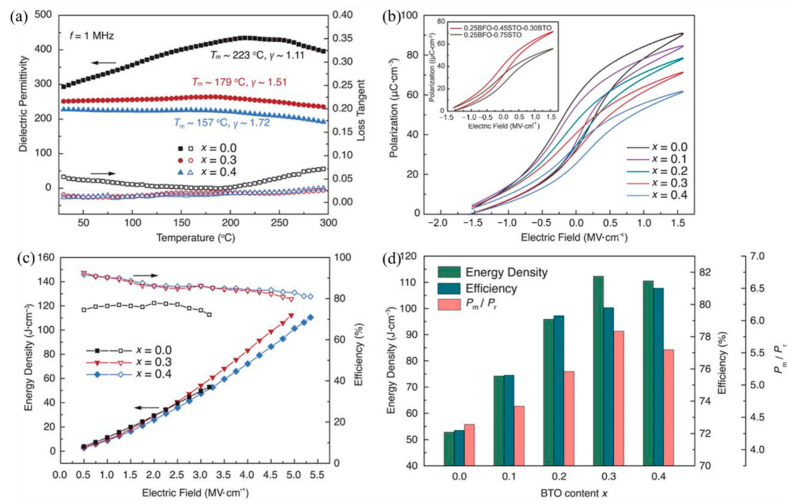
Dielectric, ferroelectric and energy storage performance of the BFBSTO films. (**a**) Temperature-dependent dielectric permittivities and loss tangents of the films at a frequency of 1 MHz. (**b**) FORC *P-**E* loops of the BFBSTO films. The inset is a comparison of the FORC loops of the x = 0.3 film (0.25BFO-0.30BTO-0.45STO) with a binary 0.25BFO-0.75STO film. (**c**) Energy density and efficiency values of the BFBSTO films with respect to applied electric fields up to their breakdown fields. (**d**) Comparison of the energy storage performance of the BFBSTO films with different BTO contents at their breakdown fields. Reprinted with permission from [94]. Copyright 2020, American Association for the Advancement of Science.

**Figure 24 molecules-26-00481-f024:**
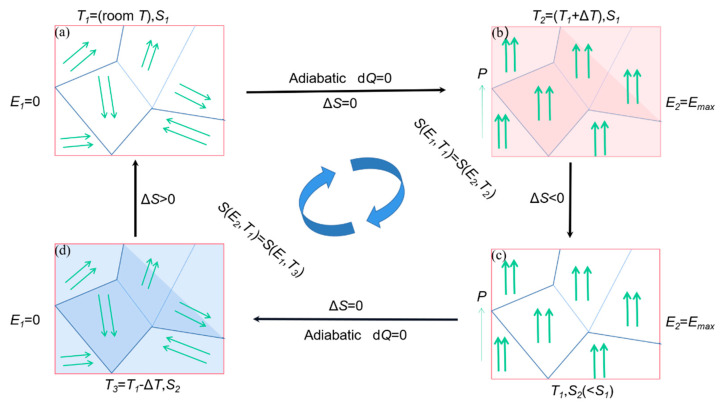
A schematic diagram for an *EC* refrigeration cycle. (**a**) The initial stage. (**b**) Adiabatic polarization stage. (**c**) Isoelectric enthalpic transfer stage. (**d**) Adiabatic depolarization stage.

**Figure 25 molecules-26-00481-f025:**
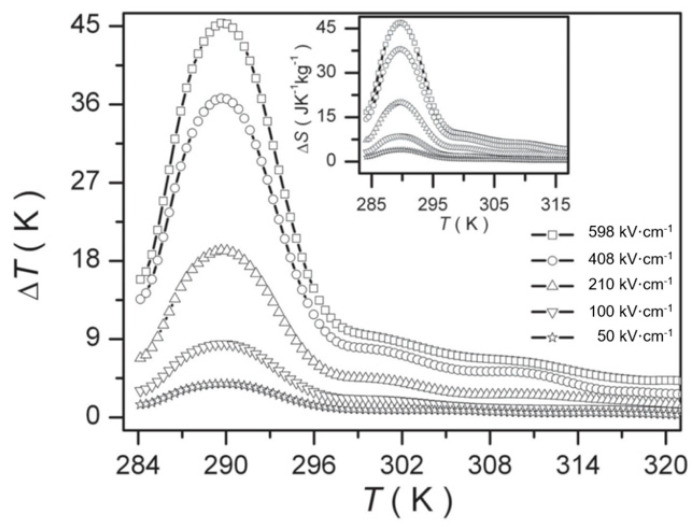
Δ *T* of PBZ film at selected electric fields, the inset is Δ*S*. Reprinted with permission from [105]. Copyright 2013 WILEY-VCH Verlag GmbH & Co.

**Figure 26 molecules-26-00481-f026:**
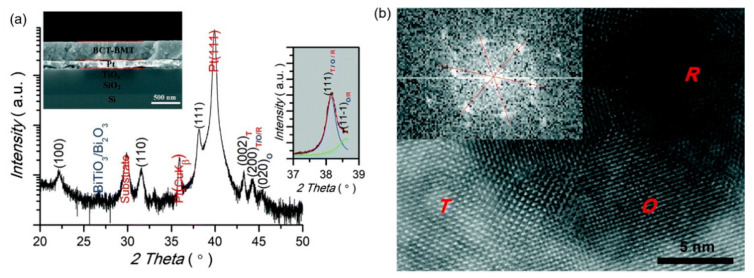
0.5(Ba_0.8_Ca_0.2_)TiO_3_-0.5Bi(Mg_0.5_Ti_0.5_)O_3_ (BCT–BMT) thin film. (**a**) XRD pattern diffraction intensity is in a logarithmic scale. A tiny amount (less than 1%) of secondary phases (BiTiO_3_ or Bi_2_O_3_) can be detected. Inset: Cross-sectional SEM image. (**b**) Atom-scale HRTEM image. Inset: The fast Fourier transform spectrum of the R region. Reprinted with permission from [92]. Copyright 2019 The Royal Society of Chemistry.

**Figure 27 molecules-26-00481-f027:**
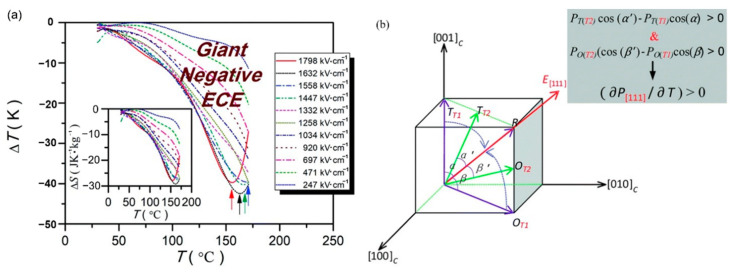
Electrocaloric effect of the BCT–BMT thin films. (**a**) ∆*T*(*T*) at selected electric fields. Inset: ∆*S*(*T*). (**b**) Phase transition diagram of tetragonal/orthorhombic to rhombohedral under the electric field. Inset: Positive pyroelectric coefficient along (111) direction. Reprinted with permission from [92]. Copyright 2019 The Royal Society of Chemistry.

**Figure 28 molecules-26-00481-f028:**
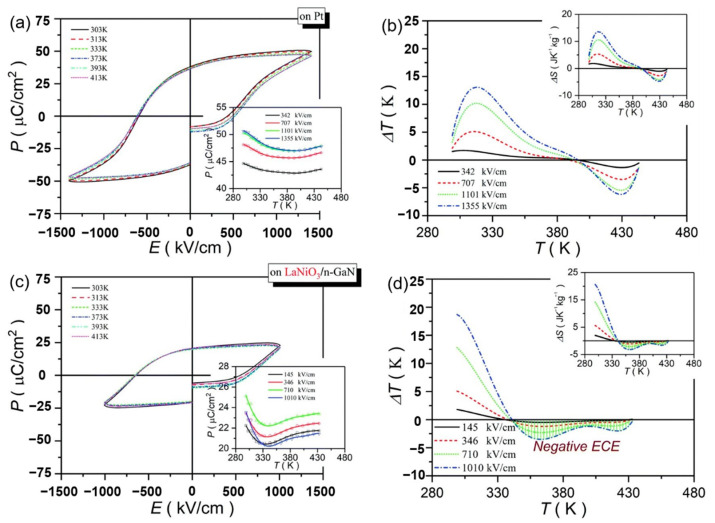
*P*–*E* loops and the corresponding *ΔT*(*T*) of the PBLZ thin films at selected temperatures at 10 kHz. (**a**,**b**) on Pt substrate. (**c**,**d**) on LaNiO_3_/n-GaN substrate. Insets in (**a**,**c**): *P*(*T*) at selected electric fields. Insets in (**b**,**d**): Δ*S*(*T*) at selected electric fields. Reprinted with permission from [106].Copyright 2019 The Royal Society of Chemistry.

**Figure 29 molecules-26-00481-f029:**
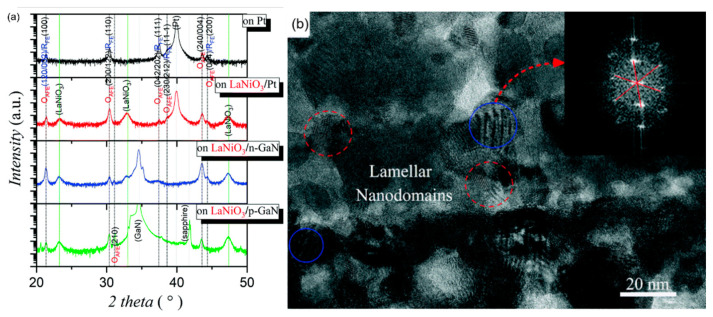
(**a**) XRD patterns of Pb_0.78_Ba_0.2_La_0.02_ZrO_3_ (PBLZ) thin films on Pt, LaNiO_3_/Pt, LaNiO_3_/p-GaN and LaNiO_3_/n-GaN substrates. FE: ferroelectric, AFE: antiferroelectric, O: orthorhombic and R: rhombohedral. (**b**) Cross-sectional TEM image of the thin film on the LaNiO_3_/Pt substrate. Inset: The fast Fourier transform spectrum of the HRTEM image of the nanodomains with a blue circle as guided by the red arrow. Reprinted with permission from [106]. Copyright 2019 The Royal Society of Chemistry.

**Figure 30 molecules-26-00481-f030:**
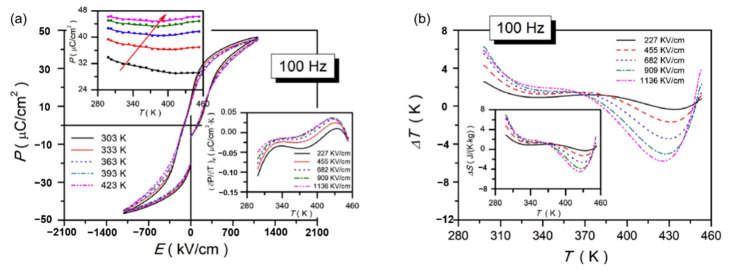
*P*–*E* loops and the corresponding *Δ**T* (*T*) of the Nb-doped PZST thin film. (**a**,**b**) at 100 Hz. Insets in (**a**): *P* (*T*) (left upper corner) and (∂*P*/∂*T*)(*T*) (right lower corner) at selected electric fields. Insets in (**b**): Δ*S* (*T*). Reprinted with permission from [107]. Copyright 2019 Elsevier Ltd. and Techna Group S.r.l.

**Table 1 molecules-26-00481-t001:** Comparison of *EC* refrigeration of different thin films.

Composition	Method	Substrate	*T* (°C)	Δ*T* (°C)	Δ*S* (J·K^−1^·kg^−1^)	Ref
0.5(Ba_0.8_Ca_0.2_)TiO_3_–0.5Bi(Mg_0.5_Ti_0.5_)O_3_	Sol-gel	Pt(111)/TiO_x_/SiO_2_/Si	163	−42.5	−25.3	[92]
P(VDF-TrFE)55/45	-	-	80	12.6	60	[95,108]
Pb_0.97_La_0.02_(Zr_0.95_Ti_0.05_)O_3_	Sol-gel	Pt(111)/Ti/ SiO_2_/Si	30	−5.0	-	[101]
PbZr_0.95_Ti_0.05_O_3_	Sol-gel	-	222	12	8	[102]
Pb_0.8_Ba_0.2_ZrO_3_	Sol-gel	Pt(111)/TiO_x_/SiO_2_/Si	17	45.3	46.9	[105]
Pb_0.78_Ba_0.2_La_0.02_ZrO_3_	Sol-gel	LaNiO_3_/GaN	-	18.7	-	[106]
Nb-doped Pb_0.99_(Zr_0.65_Sn_0.3_Ti_0.05_)_0.98_O_3_	Sol-gel	Pt(111)/TiO_x_/SiO_2_/Si(100)	20/ 152	12.3/ −5.8	13.6/ −4.5	[107]
Pb_0.97_La_0.02_(Zr_0.65_Sn_0.3_Ti_0.05_)O_3_	Sol-gel	LaNiO_3_/Pt	192	20.7	14.7	[103]
Pb(Mg_1/3_Nb_2/3_)_0.65_Ti_0.35_O_3_	Pulsed laser deposition	Si	140	31	32	[109]
0.9PbMg_1/3_Nb_2/3_O_3_–0.1PbTiO_3_	Sol-gel	Pt(111)/TiO_2_/SiO_2_/Si(100)	75	5.0	5.6	[110]
PbSc_0.5_Ta_0.5_O_3_	Sol-gel	Pt/Ti/SiO_2_ /Si	39	6.2	6.3	[111]
Pb_0.97_La_0.02_(Zr_0.75_Sn_0.18_Ti_0.07_)O_3_	Sol-gel	LaNiO_3_/ Si (100)	5	53.8	63.9	[112]
Pb_0.96_Eu_0.04_ZrO_3_	Sol-gel	Pt(111)/Ti/ SiO_2_/Si	130	−6.62	−5.42	[113]
Hf_0.5_Zr_0.5_O_2_	Atomic layer deposition (ALD)	-	175	−10.8	−10.9	[114]
Hf_0.2_Zr_0.8_O_2_	ALD	SiO_2_/Si	25	13.4	16.7	[115]
PbZr_0.53_Ti_0.47_O_3_/CoFe_2_O_4_	Pulsed laser deposition	LSCO coated (100) MgO	−91	−52.2	−94.2	[116]

## Data Availability

Not applicable.

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
