# Peer review of "Energy Storage and Electrocaloric Cooling Performance of Advanced Dielectrics"

_molecules, 2021, doi:10.3390/molecules26020481_

Round 1

Reviewer 1 Report

The new version of the menuscript looks as a completely new manuscript. It contains a huge number of changes in comparison to the previous manuscript.

In my opinion, the revised version of the manuscript still cannot be considered as a review paper. I do not find enough added value of such review. The major part of the manuscript is a loose and chaotic cluster of information collected from publications without significant systematization. This is why I cannot recommend the manuscript for publication. In the below I justify my statement.

  1. An important parameter reflecting the suitability of the material for energy storage devices is energy storage density, W. The authors cite such values from literature in the following way. Lines 366/367 - …ultrahigh energy storage density 2. 41 J/cm3”. In the line 389  W = 2.48 J/cm3 is rated as “high”. Similarly, energy storage density equal to 4,2 J/cm3  is also “high”, but 4.03 J/cm3 (line 397) – “giant”. W = 6,1 J/cm3 (line 316) is probably normal because the authors did not evaluate it. What is the criterion for such strange rating of W?
  2. A similar effect applies to energy storage efficiency. In line 367 the energy storage efficiency equal to 91.5 % is “high”, while 85.4 % (line 397) – “ultrahigh”.
  3. In the first part the references are wrongly cited with both name and surname (e.g line 72 Z. Zhang instead of Zang et al.). Only starting from p.16 a correct citation is found.
  4. All symbols of physical quantities as P, E, W, and others should be in italic not normal.
  5. In the line 39/40 one finds a phrase “symmetric center”. Crystallographers prefer “inversion center” or “center of symmetry” or “symmetry inversion center”
  6. “dielectric loss” is understood as imaginary part of permittivity/susceptibility. On the other hand “tan delta” is known as a “loss factor”. Thus the phrase in line 121 and other throughout the text shout be corrected respectively.
  7. The line 165. There is a strange phrase ”Dielectric stability..Frequency dependence..”
  8. Line 187 “relaxed ferroelectrics instead of “relaxor…”
  9. Line 177 Ther is a phrase “… improves the permittivity…” What does it mean? In what respect? Higher lower values; frequency independent?
  10. Line 195 “As shown in Fig. 7(a) …? I see hysteresis loop there.
  11. Line 256 The correct name of the theory is “Landau-Ginzburg –Devonshire”
  12. Line 257 “P” is not polarizability
  13. Line 276 “polarization intensity” – what does it mean? Physicists do not use such a name.
  14. Page 10 In Eq. 3-1 - 3-5 and Eq.3-6 two different physical quantities are used But they are represented by the same symbol “P” It is a conflict
  15. Line 306 “…thin P-E curve”; better “narrow *** loop”
  16. Line 305 “…stable change…” I do not understand it.
  17. Figure 8 “Poling” instead of “polarized”.
  18. Page 13. Starting from this page many new symbols appear, which are not explained.
    g. Wrec, Eb, Pmax and so on later.
  19. Line 350, Energy efficiency should be reported in %, according to Eq. 3-7
  20. Line 388, sodium niobate is antiferroelectric at room T, but the chapter 3-5 concerns ferroelectrics
  21. Line 386. What is Q?
  22. Line 430. Again strange quantity “polarization strength”?
  23. I randomly checked the correctness of the citations. My inspection to the Ref [32] reveal that the authors of this paper do not mention the (Ba, Sr)TiO3 system at all. On the other hand among the authors of the Ref. [99] I do not see Nan. These two cases suggest that there may be more such errors. The citations must carefully be checked.
  24. Line 500 –“‘…decreased monotonously…” There are two adjectives “monotonic” and “monotonous”. In physics/mathematics “monotonic” is preferred.
  25. Line 522, “size” cannot be measured in kV/cm

The manuscript contains a huge number of imperfections. It is hardly readable. It contains a huge number of technical information on dielectric materials. I do not see big scientific value of such paper.

Having in mind the above flaws I cannot recommend the manuscript for publication. I even doubt if it would be possible to improve it under the “major revision” option.

Author Response

Dear  Reviewer,

Thank you very much  for so detailed comments.  We have revised the paper item by item and please the attachment.

Best regards,

Prof. Wenhong Sun

Reviewer 2 Report

Unfortunately, the authors did not made exhaust improvement of the manuscript. So, the paper  still needs more corrections. Following points need to be clarified:

  • Lines 49-51, is “……and ferroelectricity are called…………with special properties are collectively called functional materials”, should be “……and ferroelectricity are collectively called functional materials”,
  • Lines 54-67, is “…properties such as various ……(with ……..the same time) and their………electrocaloric refrigeration.”, should be “…properties of various ……and their………electrocaloric refrigeration.”,
  • Line 69, is “…strength (>500) and…..”, should be “…strength (>500 kV/mm?) and…..”,
  • Figure 1, you should add information, what means “BA”,
  • Figure 2, in its caption is “Eb”, however in the Figure is “Ea”,
  • Figure 4, in its caption is “DF”, should be “dissipation factor”, a, b, c and d parts of the Figure should be removed,
  • Figure 5, its caption is completely unclear, in addition; is “hot-press”, should be “hot-pressed”,
  • Line 139, is “Polymer-ceramic”………, should be Polymer ceramic…… (you should remove quotation marks),
  • Line193, is “……..field such as. i)……”, should be “……..field such as: i)……”,
  • Line 196, is “…most critical issue. ii) …”, should be “…most critical issue, ii) …”
  • Line 197, is “….of the film. iii)…”, should be “….of the film, iii)…”,
  • Line 205, is “relaxed, should be “relaxor”,
  • Lines 222-223, is “As shown in Figure 7(a), the polarization……..the electric field E and the ….”, should be “The polarization ……..the electric field E (Figure 7) and the ….”,
  • Line 243, is “….improvement over pure STO….”, should be “….improvement over stoichiometric STO….”,
  • Lines 253-254, the sentence “The material …….is called ferroelectric material” should be removed, as the same you written above,
  • Lines 267-269, is “…..The disappearance………P(E) macroscopically.”, should be “…..The disappearance………P(E) macroscopically (Figure 7).” The next sentence should be removed,                                                                                                                                                                                                                                                                                                                                                                                                                                                                                                                                                                                                                                                                                                                             
  • Line 282, is “……..intensity (Ps).”, should be “……..intensity (Pr).”,
  • Lines 316-317, the sentence “The polarization, as shown in Figure 7.”, should be removed, as the same you written above,
  • Line 324, is “…..a large energy charge storage density.”, should be “…..a large energy storage density.”,
  • Lines 342-343, is “Ferroelectric ceramics…..reliability. However, the dielectric…..” should be “Although, ferroelectric ceramics…..reliability, the dielectric…..”,
  • Lines 347-349, is “Relaxor ferroelectrics……, as shown in Figure 6c. Relaxor ferroelectrics…..the 1950s.”, should be “Relaxor ferroelectrics, which were discovered in the 1950s, were advantages……, as shown in Figure 6c.”,
  • Line 363, is “…..adopted The …”, should be “…..adopted the …”,
  • Line 380, is “….antiferroelectric materials.”, should be “….antiferroelectric ones.”,
  • Lines 381-382, the sentence “When …….in antiferroelectric materials.”, should be removed. The next sentence should be as follows: “The polarization direction is opposite in antiferroelectrics (Figure 8a), only…….”,
  • Lines 385-386, is “….(Pb,La)(Zr,Ti)O3, ….AgNbO3, ……..etc.”, you should remove “AgNbO3” and add (K,Na)NbO3, (K,Na,Sr)(Sc,Nb)O3”,
  • At the end of part 3.4, you must at least to give two examples of antiferroelectric materials with their energy storage density values,
  • Lines 393-394 and line 400, what means Wrec, EF, Eb? You the first introduce them, so you must explain their meaning,
  • Lines 471-472, is “…..Na, K and BiFeO3……..(Na1/2Bi1/2)TiO3, etc.”, should be “…..Na, K, for example BiFeO3……..Na5Bi0.5TiO3, etc.”,
  • Lines 485, is “…..reduction, a/b site substitution,…..”, should be “…..reduction, as a result of A/B site substitution,…..”,
  • Line 507, is “……from 260~ to 350~ of pure AgNbO3.”, should be “……from ~260 to ~350ºC.”,
  • Caption of Figure 12, what means “BAN-x ceramics”, “AN”, “BAN-05”, “BAN-1””BAN-2”, “BAN-3”?,
  • Line 519, is “…..[3, 80], as shown in Figure 13.”, should be “…..[3, 80].”,
  • Line 521, is “…..size to increase Eb.”, should be “…..size to increase Eb (Figure 13).”,
  • Line 542, is“……, Pr increased and decreased continuously….”, should be “……, Pr decreased continuously….”,
  • Lines 558-559, is “Doped Sm3+……..and doped Ta5+……”, should be “Substitution of Sm3+……..and substitution of Ta5+……”,
  • Lines 567-568, you should introduce name of the material (you should make the same for caption of Figure 17),
  • Line 569, what means “SANT system”?,
  • Part 3, you should describe in detail the improving energy storage density through aging process based on Figure 18,
  • Line 656, is “…..4.4. Improving….”, is it correct?, You should also describe more about this manner of improving energy storage density,
  • Lines 670-671, is “…….volume, higher dielectric properties and ten times the energy……..”, should be “…….volume, better dielectric properties and ten times higher the energy……..”,
  • Line 672, you should make full names of “PLD”, “RF”, “CSD”,
  • Line 677, is “……0.55BZT-0.45BCT and…..”, should be “……0.55BZT-0.45BCT thin film and…..”, you should also make full names of “BZT” and “BCT”,
  • Line 681, is “……0.5…..-0.5……, should be “0.5BTZ-0.5BCT”,
  • Lines 686-687, you should make full names of “LNO” and BCZT,
  • Line 698, what means “XRR”?
  • Line 738, is “……relaxation …..”, should be “….relaxor….”,
  • Line 745, is “…..BiFeO3-BaTiO3-SrTiO3….”, should be “…….BiFeO3-BaTiO3-SrTiO3 (BFBSTO)….”,
  • Lines 787-788, is “….., the material absorbs the external temperature and causes the external temperature to decrease.”, the material absorbs the external heat, which results in external temperature decrease- you should change this sentence,
  • Line 901, is “……use in capacitors.”, should be“……use in many devices.”,
  • Line 917, is “…….BaTiO3 (BTO): BTO offers……”, should be “…….BaTiO3 (BTO) offers……”,
  • Line 959, is “……dielectric and piezoelectric materials…..”, should be“……dielectric materials…..”

The manuscript should be ones again carefully read and their deficiencies, which examples were mentioned above should be removed. The manuscript should be published in Molecules after taking into account the above comments.

Author Response

(The authors gave the same response as above.)
